# The sifting of visual information in the superior colliculus

Kyu Hyun Lee, Alvita Tran, Zeynep Turan, Markus Meister*

Division of Biology and Biological Engineering, California Institute of Technology, Pasadena, United States

**Abstract** Much of the early visual system is devoted to sifting the visual scene for the few bits of behaviorally relevant information. In the visual cortex of mammals, a hierarchical system of brain areas leads eventually to the selective encoding of important features, like faces and objects. Here, we report that a similar process occurs in the other major visual pathway, the superior colliculus. We investigate the visual response properties of collicular neurons in the awake mouse with large-scale electrophysiology. Compared to the superficial collicular layers, neuronal responses in the deeper layers become more selective for behaviorally relevant stimuli; more invariant to location of stimuli in the visual field; and more suppressed by repeated occurrence of a stimulus in the same location. The memory of familiar stimuli persists in complete absence of the visual cortex. Models of these neural computations lead to specific predictions for neural circuitry in the superior colliculus.

## Introduction

Whereas the human eye takes in about one gigabit of raw visual information every second, we end up using only a few tens of bits to guide our behavior (*Pitkow and Meister, 2014*). Of course those bits are carefully selected from the scene, and which specific bits get used depends entirely on the context and goals. All this happens in a processing time of about a tenth of a second (*Stanford et al., 2010*; *Thorpe et al., 1996*). How the visual brain sifts the onslaught of visual data for the few behaviorally relevant nuggets has been an enduring mystery. Much research in this area has focused on the primate visual system, and specifically the phenomena of invariant object recognition. For example, certain neurons in the inferotemporal cortex respond selectively to a specific individual's face regardless of its position or view angle (*Freiwald and Tsao, 2010*), or to the concept of a specific celebrity regardless of how that concept arises (*Quiroga et al., 2005*). An impressive body of theory and computational modeling has been developed to explain how this sifting for important bits from the retinal output may be implemented (*DiCarlo et al., 2012*; *Serre et al., 2007*). However, empirical evidence regarding the actual biological microcircuits has been difficult to obtain.

In rodent vision, a prominent example of visual sifting is the defensive reaction of a mouse to an approaching aerial predator (*De Franceschi et al., 2016*; *Yilmaz and Meister, 2013*). Freezing or escape can be triggered reliably by an overhead display of an expanding dark disk. Effectively, the alarm circuits in the mouse's visual system extract from the overall visual display just one or two bits of information needed to initiate action. To function properly, such an alarm system must be highly selective for the trigger feature. Indeed the mouse does not respond to expanding white disks, or to dimming dark disks, or to contracting white disks (*Yilmaz and Meister, 2013*). All these innocuous stimuli share some low-level features with the expanding dark disk, but not the overall configuration. Furthermore, the behavior is invariant to irrelevant features. For example, a mouse will freeze in response to looming stimuli presented anywhere in the upper visual field. It is unknown how this

*For correspondence:
meister@caltech.edu

invariance to location arises, and how it can coexist with high selectivity for the local stimulus features.

Recent research on rodents suggests that the visual drive for these defensive behaviors arises not in the thalamo-cortical pathway but in the superior colliculus (*Evans et al., 2018*; *Shang et al., 2018*). The superior colliculus (SC) is an evolutionarily ancient midbrain structure that mammals share with birds, fish, and amphibians (*Basso and May, 2017*; *Cang et al., 2018*). The superficial layers receive inputs from the retina and in mammals also from the visual cortex, organized in a precise retinotopic map (*Seabrook et al., 2017*). Neurons there project to the deep layers of the SC as well as other brain areas including the lateral geniculate nucleus and pulvinar. The deep layers also receive signals from other sensory modalities including hearing and touch. Neurons in the deep SC represent pre-motor signals and project broadly to many brain areas in both ascending and descending pathways. Generally speaking neural processing in the SC identifies salient points in the environment and coordinates the orienting of the animal toward or away from such locations. In the primate brain, this has been studied extensively for the special case of eye movements (*Kowler, 2011*), but the primate SC also helps control head, arm, and body movements. Furthermore, the SC contributes to a type of 'internal' orienting, namely when we direct our attention to a specific part of the scene without overt eye movements (*Krauzlis et al., 2013*).

To better understand how visual sifting proceeds in the SC we recorded spike trains simultaneously from neurons throughout all layers of this structure in the awake mouse. The set of visual displays included visual threats that reliably elicit defensive reactions, and closely related stimuli that do not. We report on three kinds of neural computations that separate behaviorally relevant from irrelevant stimuli, and we trace their emergence from the superficial to the deep layers of the SC: (1) an increasing selectivity for the threat stimulus; (2) an increasing invariance to location of that stimulus; and (3) the suppression of neural responses to a familiar stimulus. In particular, this memory of familiar stimuli is stimulus-specific, lasts for a behaviorally relevant timescale, and does not require input from the visual cortex. To explain these computations we consider several circuit models, some of which can be eliminated based on the population recordings. These results suggest how circuits of the SC can effectively distill the ecologically relevant information that guides behavior.

## Results

### Emergence of new response properties from superficial to deep layers

To track visual computations in the mouse SC, we recorded from hundreds of neurons simultaneously in all layers of the structure using multi-electrode silicon prongs (*Du et al., 2011*). The animal was head-fixed, awake, and moving on a running wheel, but not trained to perform any specific task, so we could best observe the autonomous visual functions of the SC. The recording electrodes were aimed at the dorso-medial portion of the SC, which processes stimuli in the upper visual field. Over the course of several hours, we presented a battery of visual displays, ranging from abstract stimuli like flickering checkerboards to those with ecological significance, like overhead looming disks.

In analyzing neuronal responses to these stimuli, we observed a systematic progression from the superficial layers that receive retinal input to the deep layers of the SC. To illustrate the dramatic change in how stimuli are represented, *Figure 1* compares recordings from two sample neurons, one in the superficial SC and the other in the deep SC.

The superficial neuron responded well to many different kinds of displays, such as an expanding dark disk (the classic 'looming' stimulus), a contracting white disk, a moving disk, or a dimming disk. By contrast, the deep neuron was quite selective for the looming stimulus (*Figure 1C*). Second, the superficial neuron had a small and precisely circumscribed receptive field roughly 10° in diameter. It responded only when the stimulus invaded that region. By contrast, the deep neuron responded strongly to any looming stimulus presented over a wide region that encompasses much of the visual hemifield (*Figure 1D*). Third, the superficial neuron responded reliably to repeated presentation of the identical stimulus. By contrast, the deep neuron fired only on the first presentation and failed to respond entirely to the subsequent ones (*Figure 1C*, top row).

The three characteristics found in the deep SC neuron's responses – selectivity for the looming feature, spatial invariance, and habituation to familiar stimuli – are all distinct from the signals transmitted by the retina. For example, an 'approach-sensitive' retinal ganglion cell (RGC) has been

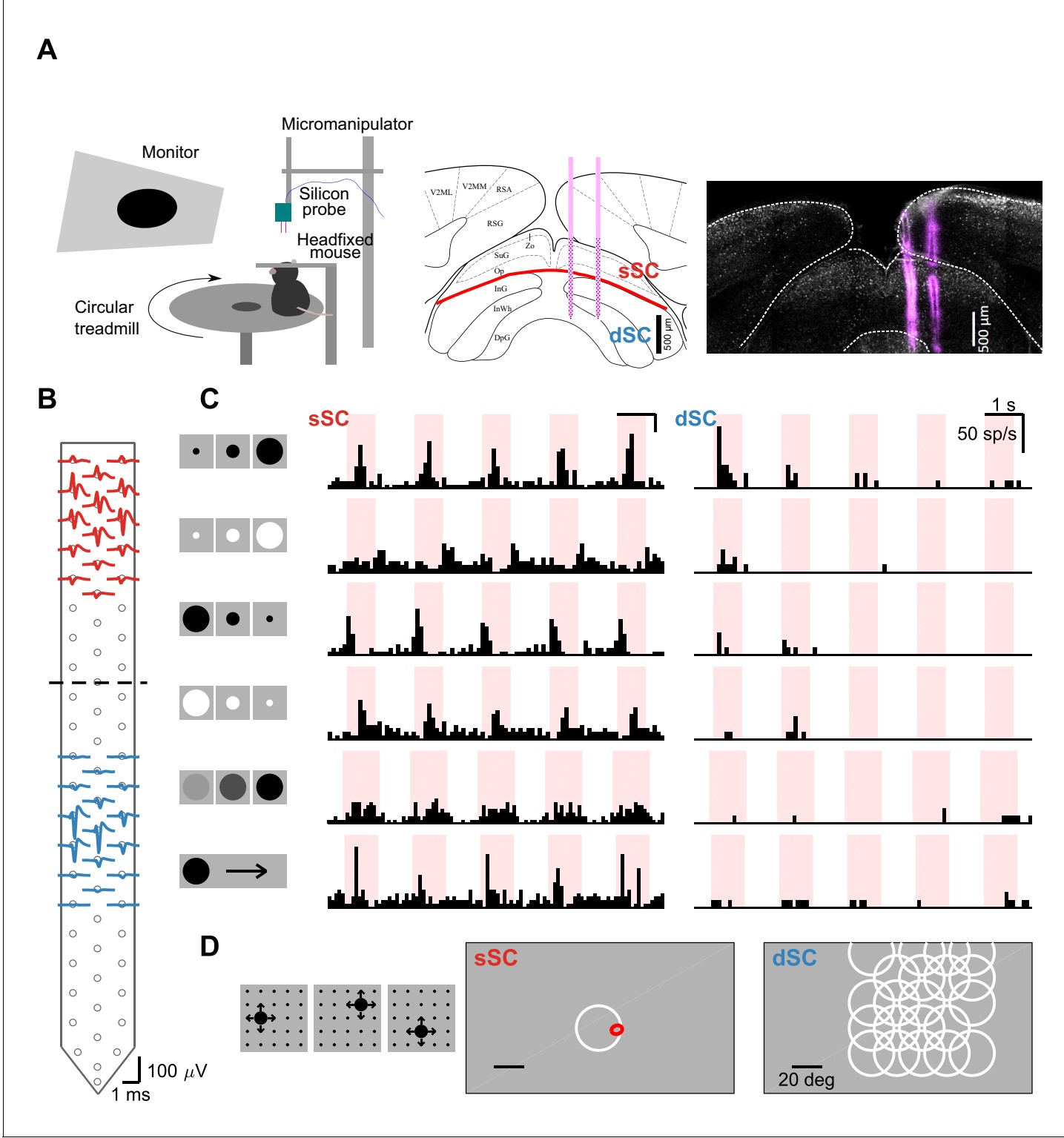

**Figure 1.** The emergence of selectivity, invariance, and stimulus-specific habituation along the depth of SC. (**A**) Left: Experimental setup. Silicon neural probes with 128 channels were implanted into the SC of a headfixed mouse viewing visual stimuli. The mouse was free to run on a circular treadmill. Middle: Diagram of a coronal section showing the anatomically defined layers of the SC (adapted from *Paxinos and Franklin, 2001*). sSC: superficial SC; dSC: deep SC. Right: Corresponding histological section recovered after neural recording, showing tracks of two electrode prongs. Magenta: DiI; white: anti-Calb1. (**B**) Extracellular spike waveforms of sample sSC (red) and dSC (blue) neurons recorded simultaneously on the silicon probe. Dots indicate the location of recording sites. Dashed line indicates boundary along the electrode array between sSC and dSC (see Materials and methods

*Figure 1 continued on next page*

*Figure 1 continued*

and *Figure 1—figure supplement 1*). (C) Response of neurons from (B) to visual stimuli. The sSC neuron (middle) responds to many types of figural stimuli (left icons: expanding black, expanding white, contracting black, contracting white, dimming, and moving black disk), whereas the dSC neuron (right) is highly selective to the expanding black disk. The sSC neuron responds robustly to every trial, whereas the dSC neuron responds primarily to the first presentation. (D) In an experiment in which looming stimuli appear from many locations (left), the sSC neuron from (B) (middle) is driven only by stimuli that cross its receptive field, whereas the dSC neuron from (B) (right) responds to stimuli placed at many more locations. White: final size of looming stimuli that elicited significant response from the cell; red: one standard deviation outline of spatial receptive field recovered by spike-triggered average method.

The online version of this article includes the following figure supplement(s) for figure 1:

**Figure supplement 1.** Histological and electrophysiological identification of SC layers.

---

reported in the mouse retina (*Münch et al., 2009*), but later studies have found that it is actually the Off-transient alpha cell (*Roska and Meister, 2014*) that responds to many other Off-type stimuli in addition to the looming stimulus (*Krieger et al., 2017*). RGCs also have local receptive fields ranging up to 10° at most (*Krieger et al., 2017*), which can be readily mapped with white noise stimuli such as flickering checkerboards or bars (*Zhang et al., 2012*). Finally, although RGCs show complex adaptation properties, the timescale of adaptation is typically on the order of 0.1 -10 (*Baccus and Meister, 2002*; *Wark et al., 2009*), whereas the habituation we find in the deep SC lasts on the order of minutes. In the following sections, we elaborate on these response properties and how they may arise in the circuitry of the SC.

## Selectivity for looming stimuli

In an attempt to measure the visual receptive fields of all the recorded neurons, we applied a flickering checkerboard stimulus and then computed the spike-triggered average (STA) stimulus (*Chichilnisky, 2001*). This is a common procedure that works well for retinal ganglion cells and neurons in the early stages of visual cortex (*Meister et al., 1994*; *Niell and Stryker, 2008*). In the superficial SC, the STA analysis yielded linear receptive fields that resembled those of retinal ganglion cells (*Figure 2A–B*). They were sharply defined in space, with the smallest only ~5° across. They frequently showed an antagonistic and delayed surround, and some displayed orientation- and direction-selectivity (*Feinberg and Meister, 2015*; *Inayat et al., 2015*). The great majority of these neurons (~90%) were Off cells based on the shape of the STA. By contrast, neurons in the deep SC did not produce sustained responses to the flickering checkerboard (*Figure 2A*), and thus contained no structure in the STA (*Figure 2B*). Nevertheless, these same deep SC neurons did respond strongly to certain figural stimuli, like the expanding dark disk (*Figure 2A,C–D*).

Among the various figural stimuli we tested, many neurons showed some selective tuning (*Figure 1C*, *Figure 2D*, *Figure 2—figure supplement 1*). We focus here on the comparison of an expanding dark disk with a contracting white disk (*Figure 2D*). These two stimuli are closely related in terms of local features: both contain an advancing dark edge. But the ecological interpretations are quite different: one indicates an approaching dark object and the other a receding white object. Freely moving mice take an evasive action to an expanding dark disk, but are unimpressed by a contracting white disk (*Yilmaz and Meister, 2013*). Compared to superficial SC, neurons in the deep SC indeed became more selective for the expanding dark disk (*Figure 2D*). This can be seen as sifting what is likely the most behaviorally relevant signal in the upper visual field from other distracting stimuli.

## Invariance to stimulus position

Although superficial SC neurons often had sharp receptive fields just 5-10° in diameter, deep SC neurons generally responded to stimuli over a large part of the visual field. We probed this tendency with expanding dark disks presented at many different locations, as these were the most effective stimuli in the deep SC. With increasing depth in the SC, neurons showed larger receptive fields, growing by a factor of 6 in area or more (*Figure 3A–B*). Note that the resolution of the receptive field measurement with expanding dark disks is ~15°, and as a result these receptive fields are larger than those measured by the flickering checkerboard (*Figure 2B*).

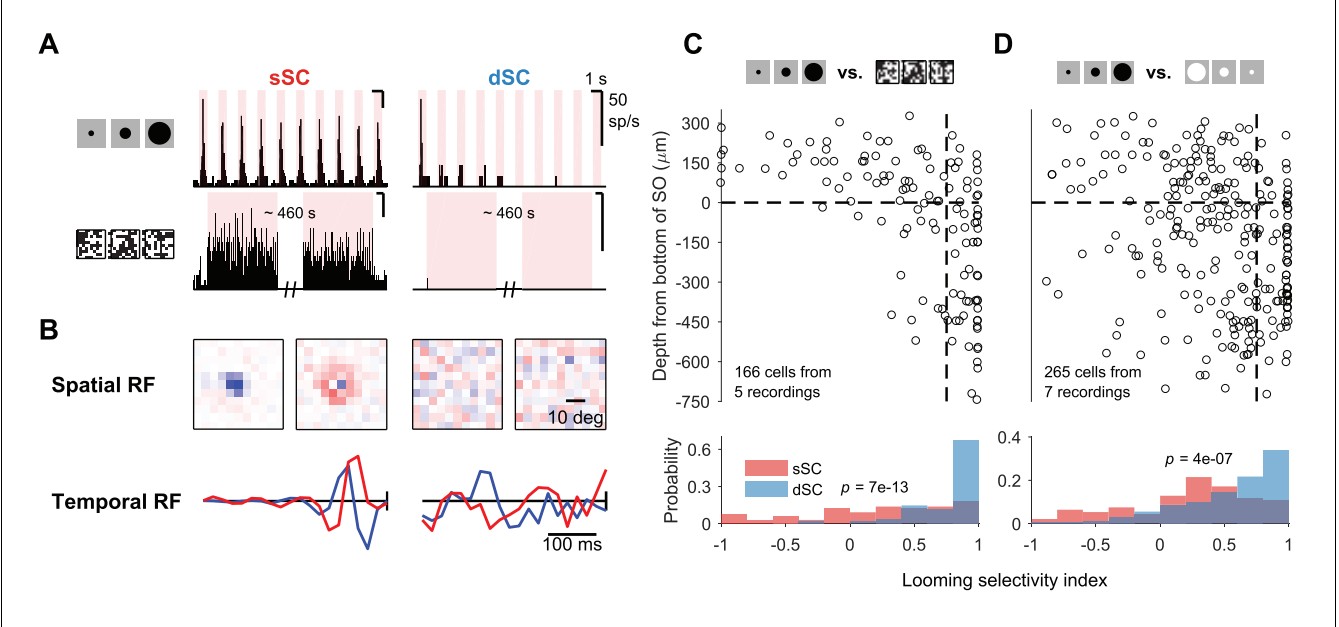

**Figure 2.** Selectivity to looming stimulus. (A) Response of sample sSC (middle) and dSC (right) neurons to looming stimulus (top) and flickering checkerboard (bottom). sSC neuron is driven strongly by both, but dSC neuron is almost completely silent to the checkerboard stimulus. (B) Spatial (top) and temporal (bottom) receptive fields of the sSC (left) and dSC (right) neurons in (A) based on spike-triggered average analysis. In each subpanel, left: spatial center; right: spatial surround; bottom blue: temporal center; bottom red: temporal surround. In the temporal RF panels, the vertical line represents the time of the spike. (C) Population summary of selectivity to looming stimulus over checkerboard stimulus along the depth of SC. Horizontal dashed line indicates the boundary between sSC and dSC. Vertical dashed line separates neurons with high selectivity index (>0.75) from others. The p-value (two-sample Kolmogorov-Smirnov test) indicates that the distributions of sSC and dSC neurons differ significantly. (D) Same as (C), comparing responses to looming stimulus and contracting white disk. Selectivity index is defined as $(r_L - r_O)/(r_L + r_O)$ where $r_L$ refers to response to looming stimulus and $r_O$ refers to response to checkerboard stimulus (C) or contracting white disk (D).

The online version of this article includes the following figure supplement(s) for figure 2:

**Figure supplement 1.** Looming selectivity over other figural stimuli.

Despite this wide spatial range, deep SC neurons responded with a remarkably short latency to looming stimuli at any location (**Figure 3A**). By the time such a neuron starts firing, the expanding dark disk has only covered a few retinal ganglion cells. In contrast, for superficial neurons the latency varied depending on the location of the expanding disk stimulus and it often exceeded the latency of deep SC neurons. (**Figure 3A**). **Figure 3C** plots this variation in the latencies across the SC depth. One possible interpretation is that a widefield neuron in the deep SC pools over many local neurons in the superficial SC, such that it becomes sensitive with the same latency at every point in its receptive field. Indeed, such an interlaminar pathway has been demonstrated previously in slice preparations (**Lee et al., 1997**; **Helms et al., 2004**). We consider this possibility more thoroughly below.

In any case, it appears that certain widefield neurons in the deep SC have solved the problem of threat detection to a large degree: they signal the looming stimulus rapidly and sensitively without false alarms from stimuli that share some low-level features but not the behavioral significance.

## Habituation to familiar stimuli

Neurons in the superficial layers generally produced a spike burst of comparable firing rate with every repeat of the stimulus (**Figure 1C**). By contrast, some neurons in the deep layers responded with a sharp burst only to the first presentation; the response to all subsequent repeats was suppressed (**Figure 4A**). The degree of habituation to repeated stimuli was greater in the deeper SC compared to the superficial SC (**Figure 4B**).

The onset of this habituation is immediate and already affects the response ~1 s later (**Figure 1C**, **Figure 4A**). The suppression then lasts for minutes: many deep SC neurons showed less than 50% recovery even after ~120 s (**Figure 4D**). While we have not measured the exact time course of

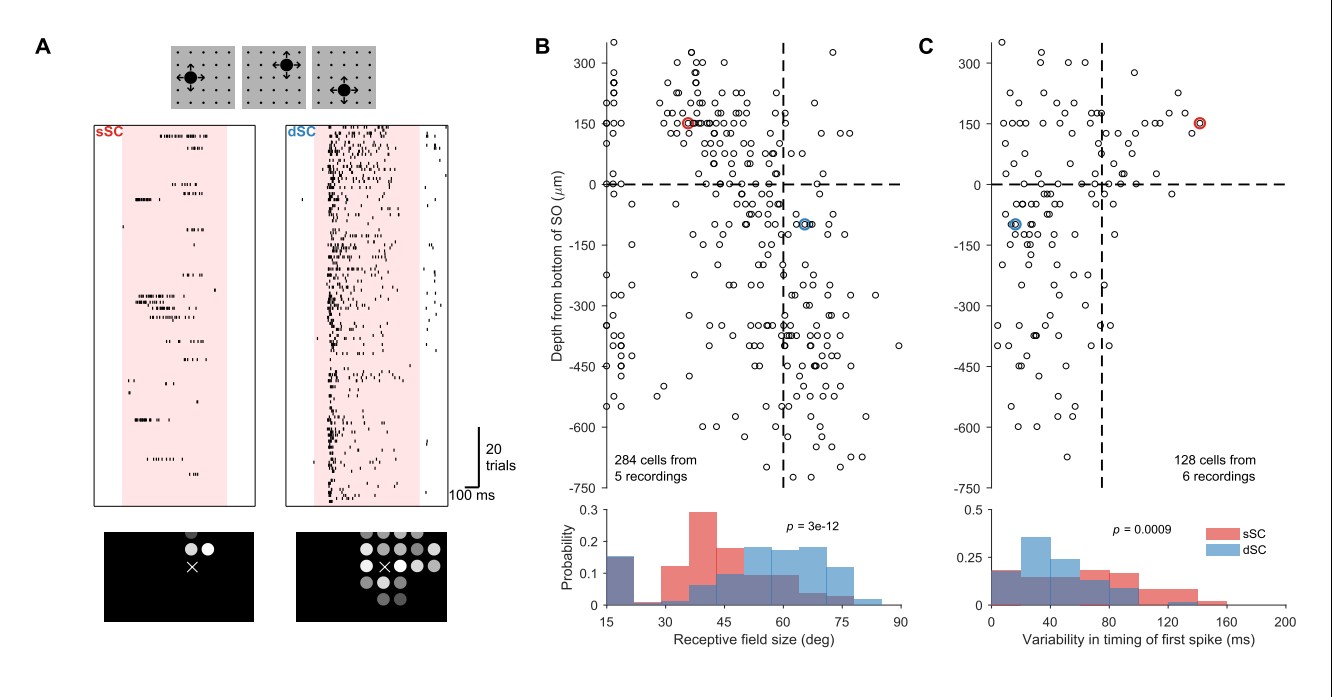

**Figure 3.** Invariance to stimulus position. (**A**) Raster plot of sample sSC (left) and dSC (right) neurons recorded simultaneously during an experiment in which looming stimuli appear randomly in one of 25 locations (small black dots in cartoon) in each trial. These locations are ~15°apart. The dSC neuron responds to many more locations than the sSC neuron and with an invariant latency. Bottom: The response amplitude at each location is reported by the brightness of the circle. X indicates a location that received no stimulus. (**B**) Population summary of receptive field size estimated from the experiment in (**A**). Vertical dashed line is at 60°. (**C**) Population summary of variability in the timing of the first spike from the experiment in (**A**). Vertical dashed line is at 75 ms. In both (**B**) and (**C**), the horizontal dashed line separates sSC and dSC. The red and blue circles denote the sSC and dSC neurons from (**A**). The p-values (two-sample Kolmogorov-Smirnov test) indicate that the distributions of sSC and dSC neurons differ significantly.

recovery, we found that the suppression was not permanent. In general, neurons recovered the full sensitivity to the first presentation when probed again about an hour later (*Figure 4—figure supplement 1*). Furthermore, the burst of spikes was not driven simply by a change in locomotor output or pupil size as a secondary consequence of the visual threat (*Figure 4—figure supplement 2*).

Remarkably, this habituation was strictly specific to the stimulus that caused the response. As reported above, widefield neurons in the deep SC can be triggered by looming disks at many different locations (*Figure 1C*, *Figure 3A*). *Figure 4C* shows the response of a single neuron to a looming stimulus whose location was chosen randomly on every trial. By comparing the sequence of responses at one location to that at another one can test whether the habituation transfers across space. As shown in the bottom left panel of *Figure 4C,*a stimulus at one location did not suppress the subsequent response of the same neuron to a stimulus at another location, even separated by as little as 15°. One interpretation is that the habituation takes place in local circuits spanning ~15° in width before their output gets pooled by the widefield neuron.

Given that the memory for familiar stimuli can last 2 min or longer, we considered whether the hippocampus or the neocortex play a role in storing this information, perhaps by modulating the gain of collicular signals through the extensive projections from visual cortex (*Zhao et al., 2014*). Thus, we repeated the experiments in a mutant mouse that lacks all the dorsal forebrain, including the hippocampus and most of the neocortex (*Kim et al., 2010*; *Figure 4—figure supplement 3*). Intriguingly, the mutant also showed long-lasting suppression of repeated stimuli in deep neurons of the SC (*Figure 4E*), to a degree that matched the suppression seen in the normal mouse (*Figure 4C* bottom right and *Figure 4D*). This is consistent with a local mechanism for habituation within the SC.

The preceding analyses of single-neuron responses suggest that the neural population deep in the SC selectively represents those bits of information that may be of immediate relevance to defensive reactions, while other aspects of the visual display get discarded. To test this directly, we

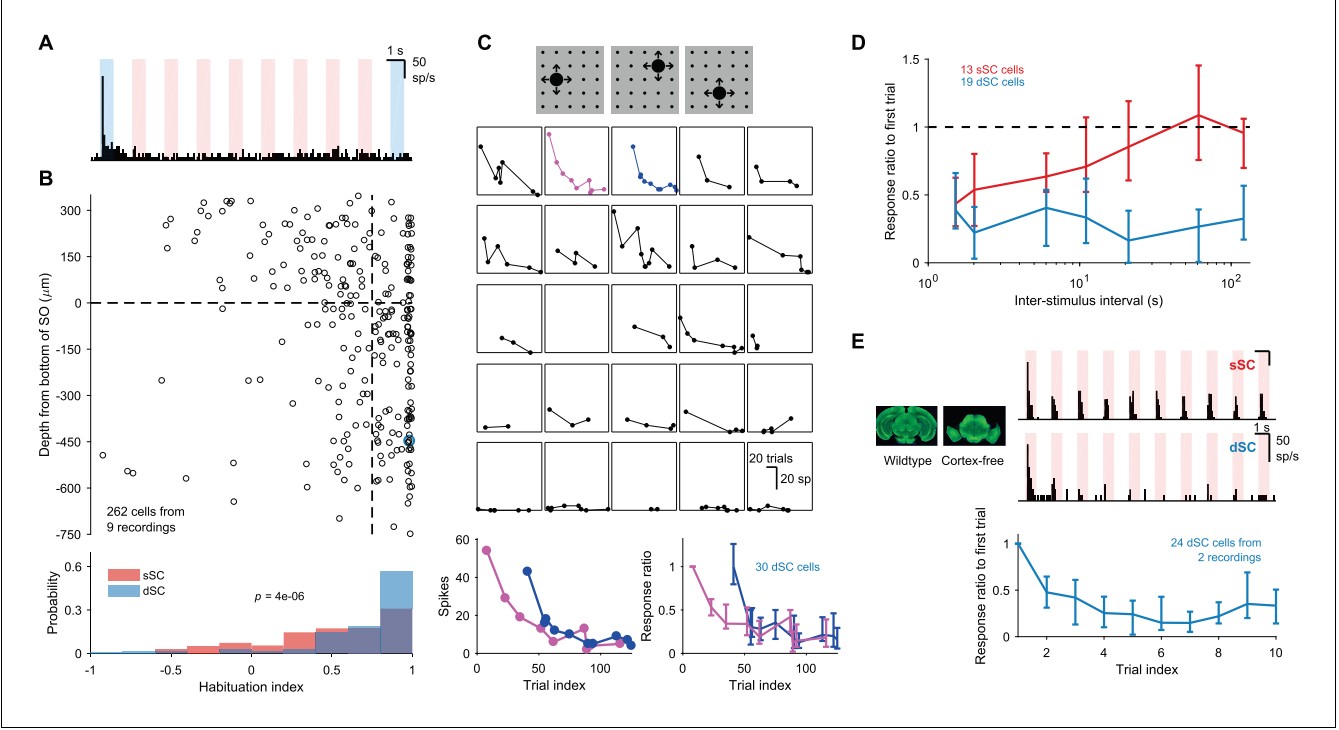

**Figure 4.** Stimulus-specific habituation. (**A**) Response of a sample dSC neuron to a series of 10 looming stimuli. The first and the 10th trials are shaded in blue. Note that this neuron has a maintained baseline firing rate, which is unchanged by the stimulus on all but the first trial. (**B**) Population summary of habituation to repeated looming stimulus. The habituation index is defined as $1 - r_1/r_{10}$ where $r_i$ refers to the number of spikes fired in in i-th trial after subtracting background activity. The horizontal dashed line separates sSC and dSC. The vertical dashed line is at 0.75. The blue circle is the sample dSC neuron from (**A**). The p-value (two-sample Kolmogorov-Smirnov test) indicates that the distributions of sSC and dSC differ significantly from each other. (**C**) Response of a sample dSC neuron to ~100 presentation of looming stimuli delivered in random sequence. Each subpanel represents response to stimuli at one of the 25 locations. Bottom left: two of the response traces from above. Even after the neuron has habituated to stimuli at one location (magenta), it responds strongly to the first stimulus at another location (blue). Bottom right: response of all dSC neurons in this recording, normalized by response to first trial of the magenta trace. Data points are medians and error bars range from 25th to 75th percentiles. (**D**) Summary of time to recover from habituation for a group of simultaneously recorded sSC and dSC neurons. Even after ~120 s, dSC neurons do not recover beyond 50% of the initial response. Data points are medians and error bars range from 25th to 75th percentiles. (**E**) Sample sSC (top right) and dSC (middle right) neurons recorded in a mutant mouse that does not develop the neocortex or the hippocampus (left). The dSC neuron in the mutant mouse also shows habituation. Bottom: population response of dSC neurons to 10 presentations of the looming stimulus, normalized by the response to the first presentation. Data points are medians and error bars range from 25th to 75th percentiles.

The online version of this article includes the following figure supplement(s) for figure 4:

**Figure supplement 1.** Suppression of a familiar stimulus is not permanent.
**Figure supplement 2.** Enhanced response to first stimulus is not a simple consequence of motor activity.
**Figure supplement 3.** A mutant mouse that lacks the neocortex and the hippocampus.

applied a linear decoder to the population vector from neurons in superficial and deep SC. From single stimulus trials, the decoder easily read out the precise location of a visual stimulus from the population in superficial SC, but much less so from neurons in deep SC (***Figure 5***, left). By contrast, the deep SC represented explicitly whether a stimulus appeared at a novel or a familiar location, whereas that information was barely available in the superficial SC (***Figure 5***, right). Of course a decoder with access to the entire history of responses could decode stimulus novelty also from the superficial SC. By contrast, in the deep SC that information is available on individual trials. In the next section, we explore how the information about stimulus history may be stored by the collicular circuit.

## A working model for circuit mechanisms of visual sifting
The microcircuitry of the SC is still poorly understood, at least compared to that of the retina. One can distinguish about 5 to 10 neuronal types based on morphology and gene expression

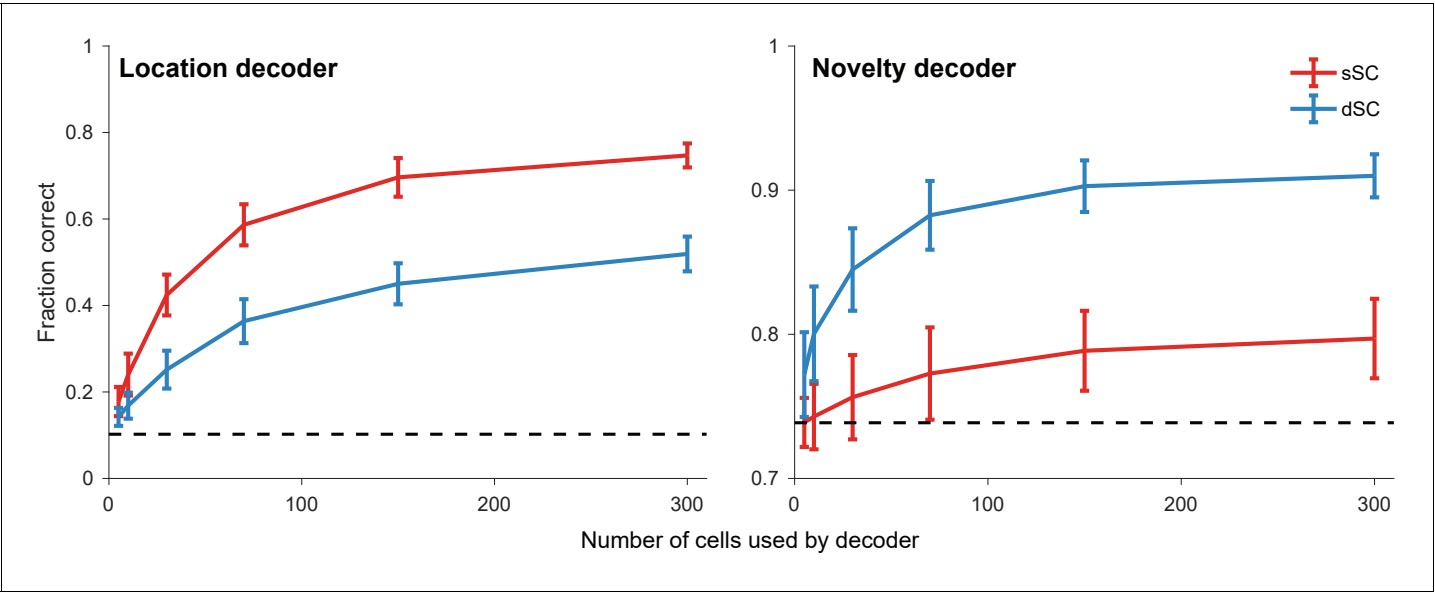

**Figure 5.** Population decoding of distinct stimulus features. Linear decoders were trained with simultaneously recorded sSC and dSC neurons to predict location (left) and novelty (i.e. whether the stimulus has appeared at a location for the first time) (right) of stimuli in the experiment described in *Figure 3*. Dashed line: chance performance; error bars: one standard deviation across different subsamples of cells.

(*Byun et al., 2016*; *Gale and Murphy, 2014*), but their synaptic connectivity is largely unknown. Furthermore, the SC interacts through long-range connections with other brain regions, notably the visual cortex (*Seabrook et al., 2017*). Nevertheless, it is useful to consider what circuit mechanisms may produce the observed visual responses of SC neurons. The functional evidence we have gathered here makes some potential explanations unlikely, and supports others as a guide in future studies of synaptic connectivity. Here, we focus on explaining three aspects of visual processing encountered in some deep SC neurons: the selectivity for looming stimuli, the invariance to spatial location, and the long-lasting stimulus-specific habituation. None of these phenomena occur in responses of retinal ganglion cells, and thus they must arise from post-retinal circuitry.

One circuit model that accounts for all the observed effects is shown in *Figure 6A* ('the working model'). It starts with input signals from retinal ganglion cells. Those are combined to produce neurons selective for a local looming stimulus. The outputs of many such local looming detectors are pooled to produce neurons with widefield sensitivity and position invariance. Finally, the input synapses to those widefield neurons undergo a short-term synaptic depression that accounts for the stimulus-selective habituation.

To simulate the function of this circuit we modeled each of the neurons as a Linear-Nonlinear element (*Chichilnisky, 2001*), and the synapses according to a widely used formalism for short-term plasticity (*Tsodyks et al., 1998*). This model correctly recapitulates the preference for looming over other stimuli (*Figure 6E*); the position invariance; and the habituation to familiar stimuli (*Figure 6F*). It even accounts for detailed dynamics of the looming response in deep neurons, such as the short latency (*Figure 3A*) and the rapid quenching of the response caused by synaptic depression (*Figure 1C*, *Figure 4A*).

While a successful circuit model seems promising, one learns something useful only from comparing different explanations. Here, we consider several alternative microcircuits to account for the looming selectivity and the stimulus-selective habituation.

The working model (*Figure 6A*) builds on local looming-selective neurons. We encountered multiple cells in the superficial SC that match this profile: a local receptive field, looming selectivity, and little habituation (*Figure 6—figure supplement 1*). In the working model, this selectivity is achieved by combining signals from retinal ganglion cells (RGCs) with different dynamics: excitation from a fast and transient Off-cell forms the receptive field center, and inhibition from slow and sustained Off-cells forms the surround. Since RGCs are excitatory, the inhibition requires interneurons in the

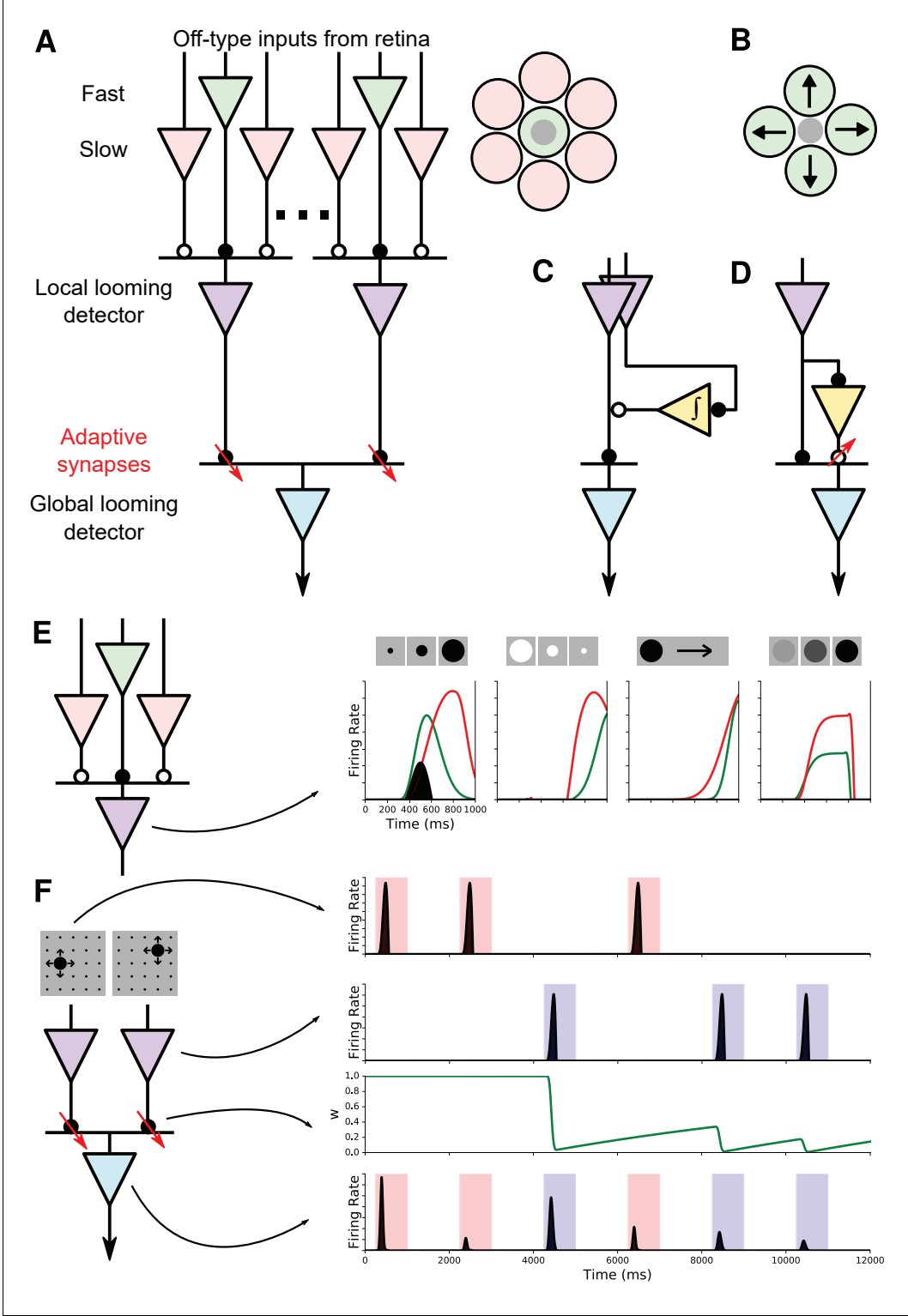

**Figure 6.** Model of selectivity, invariance, and stimulus-specific habituation. (**A**) The 'working model' of how selectivity, invariance, and habituation arise in the dSC. Looming selectivity is generated by combining fast and slow Off-type retinal inputs (green and pink) in the local looming detector (purple) in sSC. Inset on right shows spatial layout of these inputs. Invariance arises from pooling these local looming detectors to a single global looming detector (cyan) in the deep layers. The stimulus-specific habituation is achieved by synapses that undergo activity-dependent short-term depression (red downward arrows). Solid circles: excitation; open circles: inhibition.

*Figure 6 continued on next page*

*Figure 6 continued*

(B) An alternative model of looming selectivity based on pooling directionally tuned inputs. (C, D) Alternative models of stimulus-specific habituation: the same input as the excitation drives a persistent inhibition (C) or a facilitating inhibitory synapse (D). (E) Simulation of responses to various figural stimuli. Green: excitation from center; red: inhibition from surround; shaded black: net response. (F) Simulation of stimulus-specific habituation. Each local looming detector connects to the global looming detector with a synapse whose strength $w$ decays rapidly and recovers slowly.

The online version of this article includes the following figure supplement(s) for figure 6:

**Figure supplement 1.** A putative local looming detector.

SC, and the slow dynamics of the surround may well result from filtering by those interneurons. In either case, the concentric organization of fast excitation and slow inhibition produces selectivity for looming over contracting white or moving or dimming stimuli (*Figure 6E*).

As an alternative explanation, could the looming selectivity already originate in RGCs? As we noted previously, the 'approach-sensitive' Off-cell that has been previously reported (*Münch et al., 2009*) is now known to correspond to the Off-transient alpha cell (*Roska and Meister, 2014*) which – while sensitive to looming stimuli – responds equally well to dimming and flashing spots (*Krieger et al., 2017*). Therefore, these RGCs do not qualify as the local looming detectors.

Another possibility is that looming selectivity results from a radial organization of direction-selective (DS) neurons, each of which reports a segment of the advancing dark edge (*Figure 6B*). Supposing those DS inputs come from the retina, the only candidates are the On-Off DS RGCs (*Sanes and Masland, 2015*), which would be equally sensitive to On edges. Thus, the looming detectors in the SC should respond to an expanding white disk as well, unlike what we observed (*Figure 1C*). If, on the other hand, the DS signals are generated de novo in the SC, one would expect to find such interneurons with all possible preferred directions. Instead, DS neurons in a given region of the superficial SC have a strong bias for just one or two preferred directions (*de Malmazet et al., 2018*). In summary, both of the considered alternative microcircuits for looming selectivity seem unlikely given the available evidence.

In the working model (*Figure 6A*), the stimulus-selective habituation is produced by activity-dependent depression of the synapses that convey the local looming signals to the widefield neuron. A plausible alternative mechanism would involve long-lasting inhibition of the looming detector from a neuron triggered by that same local stimulus (*Figure 6C*). This neuron would need to exhibit a sustained activity following a single stimulus. In our database of collicular recordings, we never encountered a neuron that matches this description. Another possibility is that local looming detectors – in addition to exciting the widefield neuron – also inhibit it via an interneuron (*Figure 6D*). Then the long-lasting habituation could be explained by the potentiation of the inhibitory synapse, rather than depression of the excitatory synapse. In that case, one might expect that repeated looming stimuli should produce a suppression of the ongoing baseline firing during later stimulus periods. We never observed such a suppression (*Figure 4A*). Instead the firing generally increased during stimulus intervals ($r_{stim}$) compared to inter-stimulus intervals ($r_{isi}$) (for 15 deep SC neurons with baseline firing > 10 spikes/s, median $r_{\text{stim}}/r_{\text{isi}}$, 25th-75th percentile range: 1.03-1.85).

In summary, several alternative explanations for the basic phenomena observed in deep SC neurons seem less likely than the working model that we propose, based on our database of extracellular recordings. We suggest that the key components of the working model in *Figure 6A*, namely the microcircuit for looming selectivity and the long-lasting synaptic depression, are fruitful targets for further investigation.

## Discussion

### Summary

The superior colliculus (SC) presents an interesting interface between purely sensory representations and pre-motor signals. Our goal here was to follow systematically how the sensory inputs from the retina get digested and filtered in the SC. As a guiding problem we chose a robust visually-triggered behavior: the defensive reaction elicited by an overhead looming stimulus. By following visual

responses of neurons from superficial to deep layers, we documented three aspects of the sifting process: (1) an increasing selectivity for the behaviorally relevant looming stimulus over other innocuous stimuli with similar low-level features (*Figure 2*); (2) an increasing invariance to other aspects of the visual display, such as the precise location of the threat stimulus (*Figure 3*); and (3) an increasing selectivity for novel over familiar stimuli (*Figure 4*). We considered how this filtering may be achieved by neural circuits and arrived at a plausible model of circuitry in the SC (*Figure 6*) that accounts for all three of the phenomena of visual sifting considered here. Moreover, several alternative circuit-level mechanisms were found to be inconsistent with the neural signals we encountered.

## Relation to earlier work

Some of the phenomena reported here have been described before in a wide range of species. A common theme is that neurons in deep SC respond over larger regions of the visual field, while retaining a preference for small stimulus features within that region (*Cynader and Berman, 1972*; *Dräger and Hubel, 1975*; *Gordon, 1973*; *Humphrey, 1968*; *Ito et al., 2017*). Also, the remarkably persistent habituation to repeated stimuli has been noted previously, even in the earliest recordings from optic tectum (*Cynader and Berman, 1972*; *Dräger and Hubel, 1975*; *Horn and Hill, 1966b*; *Lettvin et al., 1961*; *Straschill and Hoffmann, 1969*; *Woods and Frost, 1977*; *Reches and Gutfreund, 2008*). Another repeated observation is that the visual cortex appears dispensable for many aspects of visual processing in the SC (*Horn and Hill, 1966a*; *Humphrey, 1968*; *Masland et al., 1971*), although it does play a subtle modulatory role (*Zhao et al., 2014*). Looming stimuli are particularly effective for many neurons in the superficial SC (*Zhao et al., 2014*). Interestingly, the early literature missed this, perhaps because of the technical difficulty of generating an expanding dark disk with the commonly used hand-held slide projector (*Dräger and Hubel, 1975*). Our present report places these disjoint observations into a common context, namely the animal's need to distill a specific signal of ecological value from the broad range of visual stimuli. We show that SC neurons are not only sensitive to looming stimuli but also become increasingly selective in deep layers, an essential requirement for an alarm system. Further we analyze the neural code at the population level throughout this brain region, which reveals the gradual progression of stimulus filtering. Finally, we consider how these aspects of neural representation relate to neural circuitry, and evaluate alternative hypotheses for such circuits. The results allow a broader consideration of how selectivity and invariance come about in brain processing, to be pursued further below.

## Ethological significance

The present study focused on stimuli presented in the upper visual field and recordings performed from the corresponding medial region of the SC. Arguably, the most behaviorally relevant event in the upper visual field is the impending arrival of a bigger animal, such as an aerial predator. The imminent threat that these events pose may account for the profuse responses to dark looming stimuli among SC neurons in this region (*Figure 2*; *Zhao et al., 2014*). Of course the threats must be distinguished from innocuous events, like the movement of overhead foliage, or the obscuring of the sky when the animal moves under shelter. The increased selectivity to the expanding dark disk in the deeper SC can account for that selectivity (*Figure 2C*, *Figure 4B*).

How should one interpret the profound habituation to repeated stimuli in this context? For one, the habituation does not interfere with the alarm response, since the animal must react to the first occurrence of a clear looming stimulus (*Yilmaz and Meister, 2013*). If the animal escapes or freezes, and the predator approaches a second time, this is likely in a different part of the visual field, and thus unaffected by the location-specific habituation. On the other hand, if the same stimulus recurs periodically in the same location, it is more likely caused by a leaf waving in the wind. Thus, the habituation can be seen as another processing strategy to reject innocuous events from the alarm pathway.

In the lower visual field the animal has different behavioral needs, such as picking out seeds against a cluttered background, following small moving prey (*Hoy et al., 2016*), perhaps identifying urine marks (*Joesch and Meister, 2016*), and tracking optic flow. Furthermore the connectivity between SC and other brain areas seems to differ in the upper and lower visual fields (*Savage et al., 2017*). Thus, one expects a corresponding difference in the rules by which visual stimuli are sifted there, a fertile area for future study.

## Selectivity, invariance, and habituation

One remarkable phenomenon in sensory processing is the emergence of neuronal responses that are both highly selective and broadly invariant. For example, certain 'face cells' in the primate visual cortex respond selectively to one person's face regardless of the view angle, scale, or illumination (*Freiwald and Tsao, 2010*). How do these seemingly conflicting characteristics arise within sensory circuits? In the working model we propose here (*Figure 6A*) the answer is 'first selectivity then invariance'. An AND operation across input neurons with different dynamics generates a local looming-selective neuron. These pattern detectors are distributed across the visual field. Then an OR operation pools across many local pattern detectors to produce the position-invariant response of the widefield neurons (*Figure 7A*).

This seems to be the scheme in other neural systems where the circuitry is understood. For example, in the auditory brain of the barn owl certain high-order neurons are selective for a particular interaural time delay, but invariant to the frequency of the sound (*Konishi, 2003*). These appear to arise from OR pooling over lower-order neurons that are selective for the same time delay but still tuned to different frequency bands. Those delay detectors in turn arise from an AND combination of signals derived from the two ears (*Carr and Konishi, 1990*). A similar processing scheme applies in the electrolocation circuits of weakly electric fish that exhibit a jamming avoidance response sensitive to frequency but invariant to many other parameters of the electric field (*Heiligenberg, 1989*).

However, this is not the only solution. In the case of face recognition, for example, it seems implausible that the brain should build separate pattern detectors for each face at each retinal location, and then pool over those to achieve invariance. An alternative scheme produces invariance first and then selectivity (*Figure 7B*). Here, there exists only a single pattern detector. But the inputs to this neuron are routed to 'look at' different spatial locations through a shifting circuit. The sudden appearance of any stimulus could engage these shifter circuits to route the corresponding low-level visual signals into the pattern detector (*Olshausen et al., 1993*; *Ullman and Soloviev, 1999*).

The observation of habituation and its specificity to location seems to greatly favor one of these schemes. Recall that habituation is seen prominently among neurons in the deep SC that are already highly pattern-selective. In the 'selectivity first' scheme, that places the gain modulation somewhere prior to the output of the pattern detector, which is the last spatially localized signal (*Figure 7A*). By contrast, the 'invariance first' scheme requires the gain modulation to occur in low-level visual neurons that are not yet pattern-selective (*Figure 7B*). This conflicts with our observations of neurons in the superficial SC that do not show location-specific habituation (*Figure 4B*). In summary, the robust observation of location-selective habituation in neurons of the deep SC favors a circuit model that develops selectivity before invariance.

Of course one can also envision intermediate solutions. For example, there is speculation that the visual cortex implements an alternation of AND and OR stages through a hierarchy of anatomical areas (*DiCarlo et al., 2012*; *Riesenhuber and Poggio, 1999*). Seeing that most vertebrate species do not have a neocortex, yet must solve the same problems of invariant pattern recognition, the SC seems like a promising arena for the study of high-level visual processing.

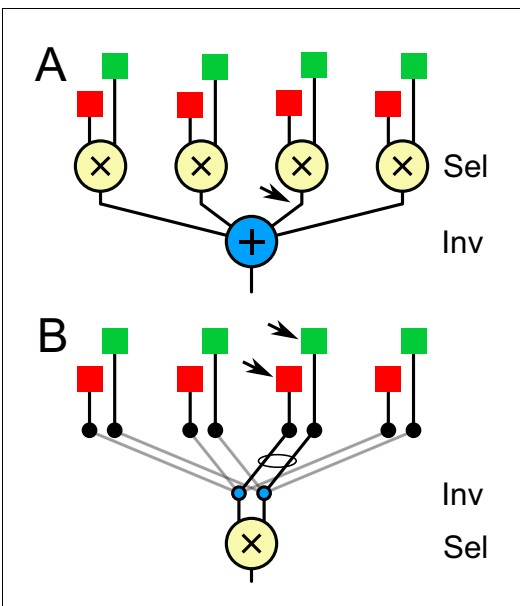

**Figure 7.** The logic of selectivity and invariance. In (**A**) feature selectivity is accomplished by combining local input signals (red and green) with AND logic (X). Then invariance arises from combining many of those feature signals with OR logic (+). In (**B**) there is only a single feature computation (X). Invariance is achieved by routing its inputs to local signals in different parts of the visual field. Arrows indicate where the stimulus-specific habituation must take place.

## Circuit mechanisms of sensory sifting

While it is tempting to suppose that the observed reduction of visual data is in fact performed within the SC, such anatomical localization is not a binding conclusion. In the extreme, the neurons of the deep SC, with their selectivity for fine spatio-temporal features and localized habituation, may simply reflect the output of a computation that occurs elsewhere. The SC interacts with many other anatomical structures (*Basso and May, 2017*; *Savage et al., 2017*), often in a reciprocal fashion. The most prominent such input, namely from the visual cortex, is likely not required for the essentials of visual sifting, based on our results with mutant mice (*Figure 4E*) and prior work with cortical silencing (*Zhao et al., 2014*) and ablation (*Horn and Hill, 1966a*; *Humphrey, 1968*). To contribute to sifting, the partner areas should retain a spatial resolution of the stimulus on the order of 10°. This constraint eliminates some small nuclei, but leaves several candidates in place, for example the thalamic area LP (*Allen et al., 2016*) and the parabigeminal nucleus. Given the position of the SC as a hub of brain pathways, it is an open question whether one can ultimately assign discrete computational functions to discrete anatomical areas.

On a finer level one may ask how the circuit models of *Figure 6* map onto neuron types in the SC. About five cell types have been distinguished in the superficial SC of mammals based on morphology alone (*Langer and Lund, 1974*; *May, 2006*), and more recent studies have connected these types to visual responses and electrophysiological properties (*Gale and Murphy, 2014*). The most compelling by their visual appearance are the so-called widefield or bottlebrush neurons. These cells have a dendritic fan that extends towards the surface of the SC and spreads out laterally to cover a large area in the retinorecipient layers. Each dendrite terminates in a bottlebrush-shaped ending, and the overall morphology is startlingly similar across birds and mammals (*Luksch et al., 1998*; *Major et al., 2000*). The widefield neurons of mammals project to the pulvinar, and the axon forms multiple collaterals in the SC that could propagate the output to the deep layers (*Basso and May, 2017*; *Major et al., 2000*).

By virtue of their broad dendritic tree these widefield neurons offer themselves as the substrate for pooling across spatial locations, as in the working model of *Figure 6A*. Two further features recommend such an identification: First, the dendrites of widefield neurons generate spikes that propagate to the soma (*Endo et al., 2008*; *Luksch et al., 2004*). In this way, the neuron truly implements an OR operation across its inputs (*Figure 7A*): when any of its inputs fire, the output will fire. Second, experiments on chick tectum showed that each dendritic input undergoes a profound synaptic depression that lasts several seconds, but does not affect the function at another dendrite (*Luksch et al., 2004*). This could account for the location-specific habituation as in the model of *Figure 6A*. However, there is some question whether this synaptic depression also happens in the mouse (*Gale and Murphy, 2016*). Also we found a substantial increase of invariance below the anatomical stratum where the widefield neurons reside (*Figure 3*).

In summary, the visual response properties of deep SC cells differ dramatically from any signal that emerges from the retina, and it is tempting to associate this transformation with the bottlebrush neuron that is shaped unlike anything in the retina. Some caution is in order, of course. The diagram of *Figure 6A* should be viewed as a conceptual scheme rather than an explicit circuit with one-to-one corresponding real neurons. Perhaps the selectivity and invariance are accomplished in multiple stages, or with the contribution of other brain areas. Or the local looming detectors may be nonlinear dendrites, and ion channels with long-lasting inactivation (*Ulbricht, 2005*) may play the role of depressing synapses. The increasing availability of genetic handles for cell types in the SC (*Byun et al., 2016*; *Gale and Murphy, 2014*) should help in cracking some of these microcircuits.

## Materials and methods

### Mouse, surgery, neural recording and spike sorting

We used C57BL/6 mice (RRID:IMSR_JAX:000664) aged 3-10 months (both males and females, Jackson Labs) for electrophysiological recordings. To prepare an animal for an experiment, we first implanted a metal headplate to the skull with a dental adhesive (3M Scotchbond) under anesthesia (2% isoflurane). After three days of recovery, the animal was habituated to being head-fixed on a circular treadmill for ~30 min/day for 3 days. On the day of recording, the animal was again anesthetized and a craniotomy (< 1 mm diameter) was made over the SC (0.2-0.4 anterior to lambda,

~0.5 mm lateral from midline). A small hole was made over the cerebellum to insert a silver reference wire. The craniotomy was then closed with a silicone elastomer (Kwik-Cast, WPI). After 6-8 hours of recovery, the animal was head-fixed and the craniotomy was exposed. A silicon neural probe was then lowered slowly into the brain (< 5µ m/s) and the depth from brain surface was recorded. The craniotomy was then covered with mineral oil to prevent drying of the exposed tissue. We waited a short period (15-30 min) for signals to stabilize before starting the recording. A typical recording session lasted 2-3 hours. All procedures were performed in accordance with institutional guidelines and approved by the Caltech IACUC.

The silicon neural probes were obtained from Sotiris Masmanidis (UCLA) (*Du et al., 2011*). For the majority of experiments, probe types 128A, 128AN, and 128DN were used. For data acquisition we used the RHD2000 128-channel amplifier board and the RHD2000 USB interface board (Intan). Auxiliary signals including the movement of the running wheel, timing of the stimulus, and timing of pupil video recording were collected concurrently with the neural signal. We used KiloSort (*Pachitariu et al., 2016*) for spike sorting of the data. The output of the automatic template-matching algorithm of KiloSort was visualized and manually curated on Phy (*Rossant et al., 2016*; *Rossant, 2017*).

To test if the long-lasting stimulus-specific habituation requires the neocortex and the hippocampus, we also recorded in mutant mice that developmentally lack these brain areas (*Kim et al., 2010*). These animals were bred by conditional knockout of exon 3 of Pals1 gene in cortical progenitor cells during embryonic development, achieved by crossing Pals1$^{flox/flox}$ mice with LoxP sites inserted upstream and downstream of exon three with Emx1-Cre animals (Jackson Labs, Strain 005628) expressing Cre recombinase in the cortical progenitor cells. Conditional knockout of both copies of Pals1 due to Cre-mediated recombination during development resulted in Emx1-Cre:Pals1$^{flox/flox}$ homozygous progeny used in this study (*Figure 4—figure supplement 3*).

## Behavioral measures

The animal's pupil diameter and locomotion on the circular treadmill were recorded along with the neural signals. The animals were not trained in any particular task and varied in their tendency to run on the treadmill. When looming stimuli were presented, the animals sometimes reacted by stopping (if the stimulus had arrived during a movement bout) or showing an increase in the pupil size (*Figure 4—figure supplement 2C*), but no characteristic behavioral output was consistently observed. However, we could rule out the possibility that the strong response of deep SC neurons to the first presentation of the looming stimulus is a simple consequence of motor output or change in pupil size, as they were usually not modulated by these factors in the absence of the looming stimulus (*Figure 4—figure supplement 2*). We also tracked the position of the pupil to monitor the eye movements. In many cases, the eyes were very stable, as demonstrated by the sharp (~5°), circular receptive fields we recovered (*Figure 2B*) in the superficial SC by spike-triggered average analysis.

## Post-hoc identification of the recorded brain area

Prior to implanting into the brain, the tip of the silicon probe was covered with fluorescent lipophilic dye (DiD or DiI, Invitrogen). Immediately after recording, the animal was anesthetized and perfused with saline and 4% PFA. The brain was harvested and fixed with 4% PFA (Electron Microscopy Sciences) for 24-48 hours at 4°C, after which it was sectioned coronally at 100 µm thickness with a vibratome (Leica). The sections were then stained with anti-Calb1 antibody (Swant, CB-38a, 1:1000 dilution), which has been previously reported to label the superficial gray layers of the SC (*Rousso et al., 2014*). Following secondary antibody staining (AlexaFluor 488, donkey-anti-rabbit, 1:1000 dilution), sections were mounted with Vecta-Shield:DAPI and imaged using a confocal microscope (LSM800, Zeiss). From this we could estimate the location of the probe relative to SC layers (*Figure 1—figure supplement 1*). This histology-based method of localizing the probe relative the SC layers was complemented with current source density (CSD) analysis. First, the raw, broadband recording was low-pass filtered (150 Hz cutoff) to isolate the LFP band. Then the Laplacian of a column of spatially contiguous electrodes was computed and smoothed with a Gaussian kernel. This revealed a series of current sources and sinks in response to visual stimulation (*Figure 1—figure supplement 1*). By comparing this CSD analysis to the histological localization, we confirmed the results from *Stitt et al. (2013)* that the inflection point between the current source and sink marks the

bottom of the superficial gray layer (SGS). We then defined the boundary between the superficial and deep layers as 100 µm below the inflection point (corresponding to 0 depth in *Figure 2C–D*, *Figure 3B–C*, and *Figure 4B*) to account for the thickness of the optic layer.

## Stimuli

Visual stimuli were programmed using the Psychtoolbox (*Brainard, 1997*; *Kleiner et al., 2007*) package in MATLAB (Mathworks) and presented on a gamma-corrected monitor (IPS231, LG) at baseline luminance of ~25 cd/m$^2$. The position of the monitor was slightly adjusted in each experiment such that the receptive fields of the neurons being recorded were located near the center of the monitor. Usually this was at ~35° in elevation and ~45° in azimuth (to the left) from the rostro-caudal axis of the animal. The monitor was located 15-20 cm from the animal and covered ~120°of the horizontal field of view. The visual stimuli were synchronized to the neural recording by using a photodiode to send timing pulses from the monitor to the data acquisition board.

Before presenting the stimuli, we used a small flickering spot to map the part of the monitor that elicited strong neural responses ('response zone'). Figural stimuli were then presented at these locations. The following is a description of each stimulus type during the stimulus period.

### Definition of stimulus period

Throughout this report, the periods during which the stimulus was presented on the screen are called stimulus periods and are marked as pink sections in the PSTHs. Outside the pink sections, the screen was uniformly gray.

### Looming stimulus

The looming stimulus expanded from 0° to ~30° at a linear expansion rate of ~30-60° /s and then remained stationary for another 250 ms before disappearing. It was presented at the full contrast achievable by the monitor. This repeated for 5-10 trials at the same location. The inter-stimulus interval was 1-3 s, except when the time to recover from habituation was explicitly tested (*Figure 4D*).

### Other figural stimuli

The contracting black, expanding white, and contracting white disks were presented with similar parameters as the looming stimulus. The stationary period of 250 ms was always at the end of the expansion or the contraction. The dimming and the moving dark disks were the same size as the final size of the looming stimulus. The rate of change in contrast of the dimming disk and the trajectory and the speed of the moving dark disk were set such that they had roughly the same duration as the looming stimulus. The moving dark disk traveled at ~40-70° /s, with the response zone in the middle of the trajectory. Several different movement directions were tried.

### Flickering checkerboard

During the flickering checkerboard stimulus, the entire screen was divided into square checkers (~3°) whose intensity changed randomly between black and white in every frame at a refresh rate of 60 Hz. The duration ranged from 300 to 600 s, but often 300 s was enough for the spike-triggered average analysis.

### Random loom

In the 'random loom' experiment, 25 locations (in a 5 × 5 grid) around the response zone were selected, with ~15° between adjacent locations (measured from center to center). In each trial, one looming stimulus was presented in one of these locations with the parameters described above. The sequence of stimulus locations was determined with a pseudorandom number generator. The inter-stimulus interval was 3 s and ~60-120 trials were presented in total.

## Analysis

The progression of visual response properties with depth in the SC was discovered in early exploratory experiments. A subsequent round of recordings was performed to validate the initial observations. The present manuscript analyzes data from only these replication experiments. All analysis scripts were written in MATLAB R2016b (Mathworks) unless otherwise noted.

## Definition of neural response and background activity

Throughout our analysis, the neural response is defined as the number of spikes that a neuron fired during the stimulus period (as described above). Some neurons had a maintained baseline firing rate. The background activity is defined as the expected number of spikes contributed by the baseline firing rate during the stimulus period. To compute this, we estimated the baseline firing rate by counting the spikes fired during the ~5-10 s-long period just preceding the stimulus and dividing by length of this period. We then multiplied this by the stimulus period to get the background activity. The background activity was used to test if the neural response was visually driven (see below).

## Identification of visually responsive neurons

Many of the recorded neurons had no clear response to visual stimuli. In a typical neurophysiology experiment, visually responsive neurons can be separated from others by presenting the stimulus many times and choosing only those that respond consistently across repetitions. In our experiments, we did not have the luxury of repeating the stimuli, as many neurons (esp. in the deep SC) showed significant habituation after just a single presentation (*Figure 4A–B*). To identify visually responsive neurons from single trials, we instead used a statistical method. First, we computed the neural response and the background activity (see above). We then computed a p-value for the neural response based on a Poisson noise model whose mean was the background activity. If the p-value was less than the pre-set cutoff of 0.005, we considered the response to be visually driven. In cases where the background activity was very low (< 1 spike), the mean of the Poisson model was set to one so that chance firing of 1–2 spikes during the stimulus period would not be considered as a visual response.

This significance criterion was used to select neurons to include in the analysis shown in *Figure 2*, *Figure 3* and *Figure 4* (see below). When computing quantities of interest (e.g. selectivity index), we first subtracted the background activity from the neural response. In *Figure 3*, the analysis required identification of significant responses from a series of stimulus presentations. To compensate for this multiple comparison, we applied a Bonferroni correction by dividing the p-value cutoff by the number of stimulus presentations.

## Receptive field analysis with flickering checkerboard

To measure the spatio-temporal receptive field (*Figure 2B*), we computed the spike-triggered average stimulus (STA) with the neural response to the flickering checkerboard (*Meister et al., 1994*). In many neurons that had a strong STA, we could separate the center and the surround of the receptive field by performing singular value decomposition (SVD) on the STA (*Wolfe and Palmer, 1998*). SVD expresses the spatio-temporal STA as a sum of terms, each of which is a product of a purely spatial and a purely temporal function. The terms are ordered by decreasing contribution to the overall variance in the STA data. We found that often the first term corresponded to the spatial and temporal profile of the center, and the second term to those of the surround.

## Stimulus selectivity

To analyze the selectivity to a looming stimulus over other stimuli (*Figure 2C,D*), we computed the looming selectivity index defined as $(r_L - r_O)/(r_L + r_O)$ with $r = r' - \mu$, where $r'$ refers to the number of spikes that a neuron fired during the first presentation of the stimulus, $\mu$ refers to the number of spikes expected during the stimulus period from the neuron's baseline firing rate, and the subscripts $L$ and $O$ refer to the looming stimulus and another stimulus (e.g. contracting white disk), respectively. For the comparison to flickering checkerboard (*Figure 2C*), $r_O = \langle r_C \rangle t_L - \mu$, where $\langle r_C \rangle$ is the average firing rate of the neuron during flickering checkerboard and $t_L$ is the duration of the looming stimulus. Only neurons that were significantly responsive to either of the two stimuli being compared based on the Poisson significance criterion outlined above were included in the analysis.

## Position invariance

To analyze the invariance to stimulus location (*Figure 3B*), we estimated the receptive field of recorded neurons from the results of the 'random loom' experiment in which looming stimuli appeared randomly at one of 25 locations (5 × 5 grid) in each presentation. The looming stimulus was chosen because unlike the checkerboard stimulus, it reliably drove both sSC and dSC neurons.

First, we defined the function $r(\mathbf{x})$ that specifies the maximum response (in spikes) of a neuron to a stimulus at location $\mathbf{x} = (x_1, x_2)$. Then we (1) set to zero the responses that did not deviate significantly from baseline activity; and (2) subtracted the expected number of spikes during stimulus period due to baseline activity from $r(\mathbf{x})$. To capture the width of the receptive field given by the remaining responses, we computed the mean radial distance $\Delta = (\sum \|\mathbf{x} - \mathbf{c}\| r(\mathbf{x})) / \sum r(\mathbf{x})$ where $\mathbf{c} = (\sum \mathbf{x} r(\mathbf{x})) / \sum r(\mathbf{x})$ is the center of mass of the receptive field and $\| \cdot \|$ is the Euclidean norm. Then we defined the receptive field size as $2\Delta$, that is twice the mean radial distance from the center of mass. Based on this method, neurons that respond to stimuli at only a single location would have a receptive field size of zero, as $\Delta = 0$. We corrected this by adding the inter-center distance between stimuli (often ~15°) to the estimated receptive field size of all neurons, as this determined the spatial resolution of our experiment.

## Variability in response latency

In addition, we analyzed the variability of response latency during this experiment (*Figure 3C*). We defined the latency as the timing of the first spike during the stimulus period. We included only the neurons that met the following conditions: (i) background activity (as defined above) is less than 1; and (ii) shows statistically significant response to at least five trials in the random loom experiment. Condition (i) is required by our definition of latency. Condition (ii) is required because we define the variability of latency as the standard deviation of the timing of first spike, and this requires some number of samples to compute. 41 sSC and 128 dSC neurons that met condition (ii) but not (i) were discarded, and the final plot in *Figure 3C* shows 37 sSC and 70 dSC neurons. Finally, to avoid including spikes not due to visual stimulation, we required that the first spike to not occur earlier than $30\,\mathrm{ms}/$ since stimulus onset.

## Stimulus-specific habituation

To analyze the stimulus-specific habituation (*Figure 4B*), we computed the habituation index defined as $1 - r_i / r_1$ where $r_i = r'_i - \mu$ refers to the number of spikes a neuron fired in the i-th repetition of the looming stimulus ($r'_i$) after subtracting the expected number of spikes due to baseline activity ($\mu$). Analysis with $i = 4$, 7, and 10 did not yield significantly different results (*Figure 4B* uses i = 10). Only the neurons whose initial response to the looming stimuli met the significance criterion were included in the analysis.

## Statistical test

Furthermore, we tested if the empirical distributions of sSC and dSC neurons differ significantly from each other in *Figure 2C–D*, *Figure 3B–C*, and *Figure 4B*. To do so we applied the two-sample Kolmogorov-Smirnov test using the MATLAB function `kstest2`. In all cases the computed p-values were less than the pre-set cutoff of 0.005 and were reported within the figure panels.

## Recovery from habituation

To analyze the time to recover from the habituation (*Figure 4D*), a series of looming stimuli was presented at a single location with inter-stimulus intervals of 1.5, 2, 6, 11, 21, 61, and 121 s, in this order. The extent of recovery was defined as $r_i / r_1$ where $r_i = r'_i - \mu$ refers to the number of spikes a neuron fired in the i-th repetition of this series ($r'_i$) after subtracting expected number of spikes due to baseline activity ($\mu$). This was done for simultaneously recorded sSC and dSC neurons that met the significance criterion. The 25th, 50th (median), and 75th percentiles were then computed separately for sSC and dSC neurons and plotted in *Figure 4D*.

## Decoding analysis

We analyzed the population of neurons from superficial and deep SC to decode stimulus variables in the 'random loom' experiment (*Figure 5*). Specifically, we asked if the population activity contains information about the location (i.e. in which of the 25 possible locations did the stimulus appear?) and novelty (i.e. is this the first stimulus to appear at a location?) of the stimuli.

To do so, we first pooled neurons from three recordings that used similar parameters of the 'random loom' experiment. Because of retionotopy in the SC, superficial SC neurons recorded by a single shank of the silicon probe tend to have overlapping receptive fields. As a result, decoding

stimulus location from the superficial SC neurons requires sampling them throughout the retinotopic map, which is difficult to do experimentally. Working on the assumption that different parts of the map contain equivalent neural representations, we augmented the data by generating virtual neurons whose response profiles were spatially shifted copies of actual neural responses. Specifically, each copy shifted the response profile to one of the eight adjacent locations in a 3 × 3 grid with the original response profile in the center. The neurons were then divided into two groups (sSC and dSC) based on the depth of the channel with maximum waveform. This augmentation process increased the number of neurons used in this analysis from 106 (38 sSC and 68 dSC) to 963 (342 sSC and 621 dSC). Some neurons whose response profile after shifting lay outside the stimulus presentation area were discarded.

After this, the data consisted of neural response of the augmented sSC and dSC populations in each of the ~100 trials. In the case of the location decoder, the labels were multi-class and ranged from 1 to 25 (one for each stimulus location). In the case of the novelty decoder, the labels were binary (stimuli that were novel, that is the first to appear at a location, were 1; others were 0). The performance measure was the mean four-fold cross validation score. The chance performance for the location decoder is the maximum of the number of times the stimulus appeared at each of the 25 locations, divided by the total number of presentations (i.e. $\max_i\{n_i / \sum_j n_j\}$, where $n_i$ refers to the number of times the stimulus appeared at location $i$). In our data, this was roughly 10%. The chance performance for the novelty decoder is the number of non-novel presentations divided by the total number of presentations. Given that there were 25 possible locations and 100 trials, this was roughly 75%.

We then subsampled sets of 5, 10, 30, 70, 150, 300 neurons from each of the two groups and used their responses to train the location and novelty decoders. This was done with the `Logisti-cRegression` class in the scikit-learn package (*Pedregosa et al., 2011*) in Python using the following parameters: *penalty = 'l2', C = 1.0, max_iter = 5000*. This process was repeated with 100 random subsamples, and the mean and standard deviation of this ensemble were computed and plotted in *Figure 5*.

## Model

In the circuit of *Figure 6A*, we modeled each input neuron as a linear-nonlinear (LN) element. The neuron's response was calculated as

$$r(t) = N(g(t)) \tag{1}$$

where

$$g(t) = s(x,y,t) * k(x,y,t) = \int_x \int_y \int_{t'=-\infty}^{t} s(x,y,t')k(x,y,t-t')\,dt'\,dy\,dx \tag{2}$$

is the convolution of the stimulus $s$ with the spatio-temporal receptive field $k$. The receptive field $k(x,y,t)$ was parametrized as

$$k(x,y,t) = F(x,y)T(t) \tag{3}$$

$$F(x,y) = \exp\left(-\frac{x^2 + y^2}{2\sigma^2}\right) \tag{4}$$

$$T(t) = \left(\frac{t}{\tau_1}\right)^{n_1} \exp\left(-n_1(t/\tau_1 - 1)\right) - b\left(\frac{t}{\tau_2}\right)^{n_2} \exp\left(-n_2(t/\tau_2 - 1)\right) \tag{5}$$

The nonlinear transformation was a half-wave rectifier:

$$N(g) = \max(0, mg - \theta) \tag{6}$$

where $\theta$ is a threshold and $m$ is a scaling factor. The firing rate of the local looming detector neuron (LD) was computed from the difference between the responses of the center and surround neurons:

$$r_{\mathrm{LD}}(t) = N(r_{\mathrm{c}}(t) - r_{\mathrm{s}}(t)) \tag{7}$$

and the response of the widefield neuron (WF) was computed from the various local detectors as

$$r_{\mathrm{WF}}(t) = \sum_{i} w_i r_{\mathrm{LD,i}}(t) \tag{8}$$

where $w_i$ is the synaptic weight from local neuron $i$ onto the widefield neuron. We modeled the habituation in the synapse between local detectors and the widefield neuron with a differential equation of three parameters for short-term synaptic depression and recovery:

$$\frac{d}{dt}w = \frac{1-w}{\tau} - a(w - w_{\mathrm{min}})r(t) \tag{9}$$

where $\tau$ is the time constant for synaptic recovery, $a$ is a gain factor for depression, and $w_{min}$ is a floor on synaptic strength. The simulation in *Figure 6F* used $a = 1$ and $w_{\mathrm{min}} = 0$.

The temporal kernels used for the center and surround neurons feeding the local looming detector were taken from the measured receptive fields of mouse alpha retinal ganglion cells (*Krieger et al., 2017*). *Table 1* lists the parameter values chosen. We arranged local looming detectors on a grid with 15° spacing between the centers of adjacent cells.

# Acknowledgements

We thank Sotiris Masmanidis (UCLA) for kindly providing us with silicon probes. The Pals1[flox/flox] mouse strain was a gift from Seonhee Kim (Temple University) and Christopher A Walsh (Harvard Medical School). We also thank the members of Meister lab for valuable discussions and comments. This work was supported by grants to MM from the Simons Foundation (543015SPI) and the NIH (R01NS111477) and a NSF Graduate Research Fellowship to AT.

**Table 1.** Parameter values used for the model in *Figure 6*, as defined by *Equations 3, 4, 5, 6, and 9*.

**Receptive field (*Equations 3, 4 and 5*)**

| Parameter | Center | Surround |
| --- | --- | --- |
| $\sigma$ | 4.00° | 10.0° |
| $\tau_1$ | 104 ms | 84.6ms |
| $n_1$ | 2.77 | 1.24 |
| $\tau_2$ | 91.2 ms | 79.7 ms |
| $n_2$ | 3.94 | 1.87 |
| $b$ | 1.34 | 1.33 |

**Nonlinearity (*Equation 6*)**

| Parameter | Value |
| --- | --- |
| $m$ | 1 |
| $\theta$ | 0 |

**Synaptic depression (*Equation 9*)**

| Parameter | Value |
| --- | --- |
| $a$ | 1 |
| $w_{\mathrm{min}}$ | 0 |

## Additional information

### Competing interests

Markus Meister: Reviewing editor, *eLife*. The other authors declare that no competing interests exist.

### Funding

| Funder | Grant reference number | Author |
|---|---|---|
| Simons Foundation | 543015SPI | Markus Meister |
| National Science Foundation | Graduate Research Fellowship | Alvita Tran |
| National Institutes of Health | 1R01NS111477 | Markus Meister |

The funders had no role in study design, data collection and interpretation, or the decision to submit the work for publication.

### Author contributions

Kyu Hyun Lee, Conceptualization, Data curation, Software, Formal analysis, Validation, Investigation, Visualization, Methodology, Project administration; Alvita Tran, Data curation, Software, Formal analysis, Funding acquisition, Validation, Investigation, Visualization, Methodology; Zeynep Turan, Data curation, Validation, Investigation, Visualization, Methodology; Markus Meister, Conceptualization, Resources, Formal analysis, Supervision, Funding acquisition, Validation, Investigation, Visualization, Methodology, Project administration

### Author ORCIDs

Kyu Hyun Lee ⓘ https://orcid.org/0000-0001-6483-9444
Markus Meister ⓘ https://orcid.org/0000-0003-2136-6506

### Ethics

Animal experimentation: This study was performed according to approved institutional animal care and use committee (IACUC) protocols (#1656) of Caltech. All surgery was performed under isoflurane anesthesia and every effort was made to minimize suffering.

### Decision letter and Author response

Decision letter https://doi.org/10.7554/eLife.50678.sa1
Author response https://doi.org/10.7554/eLife.50678.sa2

## Additional files

### Supplementary files

• Transparent reporting form

### Data availability

The data used in the manuscript as well as the analysis codes have been made available on Caltech-DATA, under the accession number 1401 (doi:10.22002/D1.1401).

The following dataset was generated:

| Author(s) | Year | Dataset title | Dataset URL | Database and Identifier |
|---|---|---|---|---|
| Lee KH, Meister M | 2020 | Data related to Lee et al. 2020 eLife, Sifting in SC | https://doi.org/10.22002/D1.1401 | CaltechDATA, 10.22002/D1.1401 |

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
