## [Decision Letter]

**Acceptance summary:**

The paper shows clear changes in coding properties between the input and deep layers of the superior colliculus. These are interesting particularly in relation to the animal's need to retain information about relevant aspects of the input stimuli while discarded others. Similar changes in coding occur in many sensory circuits, and work like that described in the paper may help guide investigation in other circuits.

**Decision letter after peer review:**

Thank you for submitting your article "The sifting of visual information in the superior colliculus" for consideration by *eLife*. Your article has been reviewed by three peer reviewers, including Fred Rieke as the Reviewing Editor and Reviewer #1, and the evaluation has been overseen by Joshua Gold as the Senior Editor.

The reviewers have discussed the reviews with one another and the Reviewing Editor has drafted this decision to help you prepare a revised submission. While the reviewers were enthusiastic about the questions asked, several issues limited enthusiasm and prevented a complete evaluation of the work:

1) A number of key findings in the paper are supported by data from very few, in some cases single, cells. Specific examples can be found in the individual reviews appended below. The reviewers felt that they could not evaluate the significance of the work without such population analyses. This point was arrived at independently by each reviewer, and emphasized in consultation among reviewers.

2) The population analyses that are presented show substantial overlap between sSC and dSC populations. The text presents a more discrete description of this data, and as a result does not accurately convey the overlap in response properties between sSC and dSC neurons. It is important that the text accurately reflect the degree of similarity/difference in sSC and dSC responses (and overlap of population analyses such as the histograms in Figures 2 and 4).

All three reviewers agreed that adequately dealing with these points will be essential for a successful revision.

Reviewer #1:

This paper investigates how the representation of visual inputs changes as signals traverse the superior colliculus (SC). The emphasis in the paper is on how three aspects of responses to behaviorally-relevant stimuli develop: selectivity, positional invariance, and habituation. Changes in coding properties similar to those characterized here occur in other sensory circuits, and hence the exploration of these changes in the SC is likely to provide broad insight into sensory processing – a point that is made clearly in the paper. The paper is generally in good shape, but I felt that several aspects could be strengthened:

Population analyses.

Most of the main points in the paper are well supported by population analyses. A few other points would benefit from such analyses:

– spatiotemporal noise. The lack of responses to the noise stimuli in the dSC neurons compared to the sSC neurons is further evidence for an increase in selectivity. This currently is shown for single example neurons in Figure 2. Can you add population results – e.g. from something like firing rate to the noise stimulus (perhaps normalized to the response to the expanding dark spot) as a function of depth?

– SC signaling without cortex. Figure 4 shows an example of the persistence of habituation from a SC neuron recorded in a mouse lacking much of cortex. Can you analyze this in all recorded cells? Do positional invariance and selectivity for the looming stimulus persist in these mice? This is an emphasized point in the paper, so it should be explored in more detail and information about cell populations provided.

– Local looming detectors. Figure 6—figure supplement 1 shows on example of a cell with properties consistent with the local looming detectors that form the basis of the proposed model. The model would suggest that there are many such cells. Is this the only one encountered? Some discussion of the required density of these cells, and the number recorded, is important. This is particularly true because the paper argues against some alternative models based on not encountering the requisite cell types.

Description of retinal ganglion cell responses

The paper in several places argues that the recorded neurons have properties that differ from those of the retinal ganglion cell inputs to SC (e.g. subsection “Emergence of new response properties from superficial to deep layers”, last paragraph and subsection “A working model for circuit mechanisms of visual sifting”, first paragraph). It would be quite helpful to provide a summary, ideally early in the paper, of the relevant RGC response properties so that the differences noted can be evaluated/appreciated.

Model.

The model is a nice addition to the paper and provides a clear way to link the results to similar coding properties in other sensory circuits. I felt, however, that the alternative models were dismissed too strongly (e.g. Introduction, last paragraph; subsection “A working model for circuit mechanisms of visual sifting”, last paragraph; Discussion, first paragraph). For example, it would seem possible to cancel the On response of the proposed DS circuit (e.g. using signals from On DS cells). Similarly, the proposed tonically active inhibitory cell that might provide for habituation is rejected because it is not observed (but very few Local Looming Detectors appear to be observed too; see comment above). I would suggest being somewhat less emphatic in ruling these models out (but I would agree that the proposed model is the most likely of those described).

Reviewer #2:

The manuscript tells an interesting story of different visual responses to dark looming stimuli in the superficial and deep layers of the superior colliculus of the mouse. It describes that in the dSC, neurons are more selectively activated by dark looming stimuli compared to receding white stimuli, are less selective to stimulus location, and are more depressed by repeated stimulation, even at novel locations. The authors present a parsimonious three layer network model using synaptic depression to explain the location specific depression. The paper is very easy to read and presents an appealing story. Overall, the manuscript suffers somewhat from sometimes only giving anecdotal evidence and presenting differences in distributions as black-and-white differences between the sSC and dSC.

"The suppression was not permanent". Only a single example is shown. Please add some population description or statistics.

"A stimulus at one location did not suppress the subsequent response.… (Figure 4C)". To support this statement only one example neuron is shown. Please add population description and statistics.

"Remarkably the mutant showed the same long-lasting suppression.…" Again only a single example is shown. Please add quantification and statistics.

"Again, this was not observed in the data". Again, only a single example is shown, and in the image it is actually difficult to observe the ongoing baseline firing. Please add quantification to make this less anecdotal.

Reviewer #3:

The manuscript presents extracellular recordings from superficial and intermediate/deep layers of SC in awake head fixed mice able to run on a treadmill and viewing either a flickering checkerboard stimulus or one of several visual patterns (black or white looming or receding stimuli, and a moving black dot). The question is what selectivity for these visual patterns there is in SC and whether that selectivity increases in deeper layers. The question is motivated by the behavioural observations from this and other laboratories that innate defence behaviours can be elicited by some stimuli (e.g. looming black disc) but not others (e.g. receding white disc).

The fundamental observation is that most of the visual patterns seem to drive robust responses in superficial SC, but they do not drive robust responses in deeper SC, with the exception of the initial presentations of a black looming disc. In addition, responses in superficial layers are confined to small regions of visual space, but in deeper layers can be elicited from large regions of visual space. The authors show that in deeper SC the responses to black looming disc's habituate rapidly, and that this habituation is spatially specific (thus presumably in the input to these deeper SC neurons). A simple model is provided to link these observations.

The paper is easy to read and the figures present data in an accessible way. Most of the analyses presented appear appropriate and well done. However, the presented analyses do not provide enough insight and require fleshing out.

The primary claim is that visual information is sifted in the colliculus, with the deeper layers showing much more selectivity for stimuli that may be behaviourally relevant (looming black disc). However, there is so little analysis of stimulus selectivity that it is impossible to judge the veracity of the claim. The sole quantitative evidence provided is I think the comparison of responses to white receding discs and black looming discs in Figure 2C. There is a mild though significant difference in the distributions in superficial and deeper layers, but the distributions are completely overlapping. This statistical difference in one comparison is not sufficient to buttress the claim of 'sifting'. An analysis of the entire response space is needed – and if the differences remain as subtle as those in Figure 2C then there needs to be some moderation of the claims that there are strong differences in selectivity. Related to this point, response amplitude seems to be substantially reduced in the deeper layers, and one way in which selectivity could be increased is if looming black discs were more capable of driving responses throughout SC, but neurons in deeper layers have a higher threshold for activation.

The second claim is that neurons in deeper SC have much larger activation regions; this is supported by nice analyses in Figure 3, which show a progressive increase in activation area for a black looming stimulus for neurons in deeper layers. I would have liked to know whether this was specific to black looming stimuli, or also the case for other stimuli – in those neurons that responded to them, but that would require a substantial number of experiments and is not a feasible target.

The third claim is that neurons in deeper SC habituate more strongly than those in superficial SC. Figure 4B and D suggest that most neurons in both superficial and deeper layers show habituation over timescales < 10s, but that on average habituation is stronger and longer lasting in deeper layers. Perhaps the nicest result here is the spatial specificity of habituation seen in the deeper layers, which implies a 'bottom-up' desensitisation of inputs to these neurons. The decoding presented in Figure 5 lost me here – the text around the tenth paragraph of the subsection “Analysis” are not particularly informative and I am not sure how the chance levels are derived, nor how the particular choice of analysis (which includes replicating spatially shifted copies of the recorded neurons) might influence the result, nor even how many neurons in each region (superficial/deep) were in the original 3 datasets. This section needs substantial clarification.

[Editors' note: further revisions were suggested prior to acceptance, as described below.]

Thank you for resubmitting your article "The sifting of visual information in the superior colliculus" for consideration by *eLife*. Your revised article has been reviewed by three peer reviewers, including Fred Rieke as the Reviewing Editor and Reviewer #1, and the evaluation has been overseen by Joshua Gold as the Senior Editor.

The reviewers have discussed the reviews with one another and the Reviewing Editor has drafted this decision to help you prepare a revised submission. All the reviewers appreciated the revisions and felt that they made the paper considerably stronger. A few substantive issues remain – as outlined in the individual reviews below.

Reviewer #1:

This is a resubmission of a paper comparing the representation of sensory stimuli in superficial and deep layers of superior colliculus. The primary concern with the original paper was a lack of population data and corresponding statistical tests of the central conclusions. Those issues have been largely dealt with, and the paper is much stronger. I have some remaining suggestions for clarity of presentation.

Reviewer #2:

The authors responded to all my specific questions and suggestions to my satisfaction.

In the addition made in answer to one of my questions, however, it seems a small error was introduced. In the new figure panel Figure 4C bottom right with the 30 dSC cells, it is mentioned that the responses are "normalized by response to first trial of the magenta trace". This indeed seems sensible. However, the first point on the blue curve is exactly 1 (not approximately, I checked in Illustrator). Is an error made in the normalization here, or in the description, or was it so close to 1 that rounding put it exactly at 1? There are no numbers or stats given in the text, so it was not possible for me to check this point.

Reviewer #3:

The revision deals with most of the issues raised by the reviewers including myself. The paper is stronger, and presents a more compelling story. I have some remaining queries.

1) I appreciate the provision of more descriptive overviews of some of the data sets. However, the claims, in the Abstract: "neuronal responses become more selective for behaviorally relevant stimuli") and in the opening of the Discussion "an increasing selectivity for the behaviorally relevant looming stimulus over other innocuous stimuli with similar low-level features (Figure 2)") still rely on only 2 comparisons of activity (black loom vs. checkerboard, and black loom vs. receding white). Another 4 datasets are presented for the example neurons in Figure 2 but are not analysed further. I previously suggested a full analysis over this response space, which the authors argue is not feasible in a single report. But the response metric for these datasets is trivial to derive, and I cannot see why the authors cannot present (perhaps as a table in the supplementary data accompanying Figure 2) an appropriate multivariate analysis. Alternatively, the text could be modified to make clear that the statements rely on population comparisons for the two stimuli as well as a checkerboard.

2) Regardless of the what the authors think of the above, around the subsection “Stimulus selectivity”, I think the authors infer the response to the checkerboard from the mean rate across the entire presentation of the checkerboard stimulus (i.e. several minutes) while the response to the loom stimulus is the number of spikes in the first presentation (i.e. first second). If I am incorrect please clarify. If I am not then if responses habituate to the checkerboard as they do to the loom then this is not a fair comparison and a companion analysis using the same time frames for both stimuli would be appropriate.

3) The use of the word "novelty" in Figure 5 legend and in the last paragraph of the subsection “Habituation to familiar stimuli”, remains unclear to the reader and there needs to be clarity in the main text that novelty here means new location not new shape. This is a non-trivial distinction.

---

## [Author Response]

While the reviewers were enthusiastic about the questions asked, several issues limited enthusiasm and prevented a complete evaluation of the work:1) A number of key findings in the paper are supported by data from very few, in some cases single, cells. Specific examples can be found in the individual reviews appended below. The reviewers felt that they could not evaluate the significance of the work without such population analyses. This point was arrived at independently by each reviewer, and emphasized in consultation among reviewers.

As detailed below, we have added several such population analyses, in part based on new experiments.

2) The population analyses that are presented show substantial overlap between sSC and dSC populations. The text presents a more discrete description of this data, and as a result does not accurately convey the overlap in response properties between sSC and dSC neurons. It is important that the text accurately reflect the degree of similarity/difference in sSC and dSC responses (and overlap of population analyses such as the histograms in Figures 2 and 4).

As detailed below, we have made changes to the test to better reflect those relationships.

All three reviewers agreed that adequately dealing with these points will be essential for a successful revision.Reviewer #1:[…] The paper is generally in good shape, but I felt that several aspects could be strengthened:Population analyses.Most of the main points in the paper are well supported by population analyses. A few other points would benefit from such analyses:– spatiotemporal noise. The lack of responses to the noise stimuli in the dSC neurons compared to the sSC neurons is further evidence for an increase in selectivity. This currently is shown for single example neurons in Figure 2. Can you add population results – e.g. from something like firing rate to the noise stimulus (perhaps normalized to the response to the expanding dark spot) as a function of depth?

As suggested, we have added a population analysis comparing the responses of 209 cells to flickering checkerboard vs. expanding dark spot in Figure 2C. dSC neurons strongly prefer the expanding spot.

– SC signaling without cortex. Figure 4 shows an example of the persistence of habituation from a SC neuron recorded in a mouse lacking much of cortex. Can you analyze this in all recorded cells? Do positional invariance and selectivity for the looming stimulus persist in these mice? This is an emphasized point in the paper, so it should be explored in more detail and information about cell populations provided.

We performed additional experiments on acortical mutant mice and added a panel to Figure 4E with population results from dSC neurons. They display persistent habituation comparable to that in the normal mouse. We focus here on the habituation instead of stimulus selectivity or position invariance because neocortex and hippocampus have been invoked as responsible for short-term memory that might underlie the habituation. Regarding stimulus selectivity, the Discussion section cites prior work (Zhao et al., 2014) suggesting that cortex plays a minor role in the tuning of neurons in the SC.

– Local looming detectors. Figure 6—figure supplement 1 shows on example of a cell with properties consistent with the local looming detectors that form the basis of the proposed model. The model would suggest that there are many such cells. Is this the only one encountered?

On further analysis we found 6 additional putative looming detectors in the database that satisfy a strict set of conditions: (i) detected in the superficial SC; (ii) selectivity index (expanding dark vs. contracting white disk) greater than 0.75; (iii) local receptive field (< = 2 adjacent looming locations); and (iv) habituation index less than 0.5. See text changes in the fifth paragraph of the subsection “A working model for circuit mechanisms of visual sifting” and Figure 6—figure supplement 1 legend.

Some discussion of the required density of these cells, and the number recorded, is important. This is particularly true because the paper argues against some alternative models based on not encountering the requisite cell types.

It is difficult to estimate the required density of these cells, as the SC is probably involved in many behaviors beyond the looming reaction. We have weakened the claims in the text about elimination of alternative models. Also, the Discussion mentions the alternative possibility that the local looming detectors might be implemented by nonlinear dendrites within a larger neuron.

Description of retinal ganglion cell responses.The paper in several places argues that the recorded neurons have properties that differ from those of the retinal ganglion cell inputs to SC (e.g. subsection “Emergence of new response properties from superficial to deep layers”, last paragraph and subsection “A working model for circuit mechanisms of visual sifting”, first paragraph). It would be quite helpful to provide a summary, ideally early in the paper, of the relevant RGC response properties so that the differences noted can be evaluated/appreciated.

We added concrete comparisons to the relevant RGC response properties at the end of the first Results section.

Model.The model is a nice addition to the paper and provides a clear way to link the results to similar coding properties in other sensory circuits. I felt, however, that the alternative models were dismissed too strongly (e.g. Introduction, last paragraph; subsection “A working model for circuit mechanisms of visual sifting”, last paragraph; Discussion, first paragraph). For example, it would seem possible to cancel the On response of the proposed DS circuit (e.g. using signals from On DS cells). Similarly, the proposed tonically active inhibitory cell that might provide for habituation is rejected because it is not observed (but very few Local Looming Detectors appear to be observed too; see comment above). I would suggest being somewhat less emphatic in ruling these models out (but I would agree that the proposed model is the most likely of those described).

We have softened the language about eliminating the alternative models.

Reviewer #2:[…] Overall, the manuscript suffers somewhat from sometimes only giving anecdotal evidence and presenting differences in distributions as black-and-white differences between the sSC and dSC."The suppression was not permanent". Only a single example is shown. Please add some population description or statistics.

In general most or all cells recovered sensitivity to a stimulus when tested again an hour later. But we have not explored the exact time course of this recovery and simply offer Figure 4—figure supplement 1 as an example. We have amended the text to clarify this point (subsection “Habituation to familiar stimuli”, second paragraph).

"A stimulus at one location did not suppress the subsequent response.… (Figure 4C)". To support this statement only one example neuron is shown. Please add population description and statistics.

As suggested, we have added another panel to Figure 4C showing the response time course at the two locations for all 30 dSC neurons recorded in one experiment.

"Remarkably the mutant showed the same long-lasting suppression.…" Again only a single example is shown. Please add quantification and statistics.

We performed additional experiments on acortical mutant mice and added a panel to Figure 4E with population results from dSC neurons. See also response to reviewer #1 on this point.

"Again, this was not observed in the data". Again, only a single example is shown, and in the image it is actually difficult to observe the ongoing baseline firing. Please add quantification to make this less anecdotal.

We performed additional analysis of 15 dSC neurons with maintained baseline firing (>10 spikes/s), and compared the mean activity during looming stimuli to the mean activity during interstimulus intervals (*r*_stim_/*r*_isi_). We found the median of this ratio was 1.28 and the 25th-75th percentiles ranged from 1.03 to 1.85; i.e. the stimulus periods had slightly *more* activity compared to the baseline. This is inconsistent with the facilitating inhibitory synapse model, which predicts that the stimulus periods would have fewer spikes after habituation. We report this result in the revised text (subsection “A working model for circuit mechanisms of visual sifting”, eighth paragraph).

Reviewer #3:[…] The paper is easy to read and the figures present data in an accessible way. Most of the analyses presented appear appropriate and well done. However, the presented analyses do not provide enough insight and require fleshing out.The primary claim is that visual information is sifted in the colliculus, with the deeper layers showing much more selectivity for stimuli that may be behaviourally relevant (looming black disc). However, there is so little analysis of stimulus selectivity that it is impossible to judge the veracity of the claim. The sole quantitative evidence provided is I think the comparison of responses to white receding discs and black looming discs in Figure 2C. There is a mild though significant difference in the distributions in superficial and deeper layers, but the distributions are completely overlapping. This statistical difference in one comparison is not sufficient to buttress the claim of 'sifting'. An analysis of the entire response space is needed – and if the differences remain as subtle as those in Figure 2C then there needs to be some moderation of the claims that there are strong differences in selectivity.

We have added another population analysis of stimulus selectivity in the new Figure 2C, which compares the responses to black expanding disk and flickering checkerboard. As we note in the text, Figure 2D focuses on the comparison between black expanding disk and white contracting disk because they share low-level features (moving dark edge) and yet have different ecological meanings to the animal. Thus it serves as a strong test of selectivity. “An analysis of the entire response space” is not feasible in a single report.

While it is true that the distributions of selectivity index for sSC and dSC overlap (Figure 2), this is consistent with our model, which explicitly posits the existence of local looming detectors in the sSC. Finally, we word the claim carefully: The selectivity generally increases at the population level as one goes deeper into the SC. We do not assert that neurons selective to the looming stimulus are exclusively in the dSC.

Related to this point, response amplitude seems to be substantially reduced in the deeper layers, and one way in which selectivity could be increased is if looming black discs were more capable of driving responses throughout SC, but neurons in deeper layers have a higher threshold for activation.

We agree that any mechanism of stimulus selectivity requires a nonlinear computation. Our working model for looming selectivity (Figure 6) makes use of a threshold as well.

The second claim is that neurons in deeper SC have much larger activation regions; this is supported by nice analyses in Figure 3, which show a progressive increase in activation area for a black looming stimulus for neurons in deeper layers. I would have liked to know whether this was specific to black looming stimuli, or also the case for other stimuli – in those neurons that responded to them, but that would require a substantial number of experiments and is not a feasible target.

In a few studies in which the “random loom” experiment was done with contracting white disks, we found that many dSC neurons were silent or very weakly responsive. This again demonstrates their selectivity for the looming stimuli.

The third claim is that neurons in deeper SC habituate more strongly than those in superficial SC. Figure 4B and D suggest that most neurons in both superficial and deeper layers show habituation over timescales < 10s, but that on average habituation is stronger and longer lasting in deeper layers. Perhaps the nicest result here is the spatial specificity of habituation seen in the deeper layers, which implies a 'bottom-up' desensitisation of inputs to these neurons. The decoding presented in Figure 5 lost me here – the text around the tenth paragraph of the subsection “Analysis” are not particularly informative and I am not sure how the chance levels are derived, nor how the particular choice of analysis (which includes replicating spatially shifted copies of the recorded neurons) might influence the result, nor even how many neurons in each region (superficial/deep) were in the original 3 datasets. This section needs substantial clarification.

We have expanded the text in the Materials and methods section to better describe how the chance levels are derived, the assumptions underlying data augmentation, how the data augmentation was performed, and how many neurons it added to the data set (subsection “Decoding analysis”).

[Editors' note: further revisions were suggested prior to acceptance, as described below.]

Reviewer #2:The authors responded to all my specific questions and suggestions to my satisfaction.In the addition made in answer to one of my questions, however, it seems a small error was introduced. In the new figure panel Figure 4C bottom right with the 30 dSC cells, it is mentioned that the responses are "normalized by response to first trial of the magenta trace". This indeed seems sensible. However, the first point on the blue curve is exactly 1 (not approximately, I checked in Illustrator). Is an error made in the normalization here, or in the description, or was it so close to 1 that rounding put it exactly at 1? There are no numbers or stats given in the text, so it was not possible for me to check this point.

The blue trace plots the median of the response ratio for 30 deeper SC neurons normalized by the response to first trial in the magenta trace, with the error bars ranging from 25th to 75th percentiles. The median of the first data point is indeed exactly 1, because this median neuron happened to fire exactly as many spikes in the first trial of the blue trace as in the first trial of the magenta trace (11 spikes). This is not an error, just a coincidence.

Reviewer #3:The revision deals with most of the issues raised by the reviewers including myself. The paper is stronger, and presents a more compelling story. I have some remaining queries.1) I appreciate the provision of more descriptive overviews of some of the data sets. However, the claims, in the Abstract: "neuronal responses become more selective for behaviorally relevant stimuli") and in the opening of the Discussion "an increasing selectivity for the behaviorally relevant looming stimulus over other innocuous stimuli with similar low-level features (Figure 2)") still rely on only 2 comparisons of activity (black loom vs. checkerboard, and black loom vs. receding white). Another 4 datasets are presented for the example neurons in Figure 2 but are not analysed further. I previously suggested a full analysis over this response space, which the authors argue is not feasible in a single report. But the response metric for these datasets is trivial to derive, and I cannot see why the authors cannot present (perhaps as a table in the supplementary data accompanying Figure 2) an appropriate multivariate analysis. Alternatively, the text could be modified to make clear that the statements rely on population comparisons for the two stimuli as well as a checkerboard.

[Does the reviewer mean Figure 1?]

Previously the reviewer suggested “an analysis of the entire response space” which seemed like a tall order. We now understand that the request was more specifically for population-level data on the other figural stimuli in Figure 1C. We have added Figure 2—figure supplement 1, which shows histograms (similar to those at the bottom of Figure 2C and D) for the responses of superficial and deeper SC neurons to four other figural stimuli (expanding white disk, receding dark disk, moving disk, and dimming disk). Although there is a considerable overlap between superficial and deeper SC neurons in all cases, the probability of encountering neurons with high selectivity index for looming motion (> 0.75) is greater in the deeper SC.

2) Regardless of the what the authors think of the above, around the subsection “Stimulus selectivity”, I think the authors infer the response to the checkerboard from the mean rate across the entire presentation of the checkerboard stimulus (i.e. several minutes) while the response to the loom stimulus is the number of spikes in the first presentation (i.e. first second). If I am incorrect please clarify. If I am not then if responses habituate to the checkerboard as they do to the loom then this is not a fair comparison and a companion analysis using the same time frames for both stimuli would be appropriate.

The response to checkerboard in Figure 2C is indeed derived from the average firing rate throughout the stimulus (see Materials and methods for details), but we argue that this is a reasonable comparison. The reviewer raises the possibility that the responses to the checkerboard stimulus could habituate in deeper SC neurons, but habituation (in our work and in general) is defined as a reduction in response to the repeated presentation of a stimulus. The checkerboard stimulus does not repeat – in fact it presents a different visual feature (i.e. randomly drawn pattern of black and white squares) 60 times a second. There are no repeats in the same location, and thus no reason to expect a habituation similar to that observed in the repeated presentation of the expanding dark disk. The key result here (subsection “Selectivity for looming stimuli”, first paragraph) is that the deeper SC neurons respond very little to this whole ensemble of visual features, none of which is ethologically relevant.

3) The use of the word "novelty" in Figure 5 legend and in the last paragraph of the subsection “Habituation to familiar stimuli”, remains unclear to the reader and there needs to be clarity in the main text that novelty here means new location not new shape. This is a non-trivial distinction.

We have changed the text throughout this section to clarify that novelty refers to a novel location.